# Choice Bandits

**Arpit Agarwal**
University of Pennsylvania
Philadelphia, PA 19104, USA
aarpit@seas.upenn.edu

**Nicholas Johnson**[*]
University of Minnesota
Minneapolis, MN 55455, USA
njohnson@cs.umn.edu

**Shivani Agarwal**
University of Pennsylvania
Philadelphia, PA 19104, USA
ashivani@seas.upenn.edu

## Abstract

There has been much interest in recent years in the problem of dueling bandits, where on each round the learner plays a pair of arms and receives as feedback the outcome of a relative pairwise comparison between them. Here we study a natural generalization, that we term *choice bandits*, where the learner plays a set of up to $k \geq 2$ arms, and receives limited relative feedback in the form of a single multiway choice among the pulled arms, drawn from an underlying multiway choice model. We study choice bandits under a very general class of choice models that is characterized by the existence of a unique 'best' arm (which we term generalized Condorcet winner), and includes as special cases the well-studied multinomial logit (MNL) and multinomial probit (MNP) choice models, and more generally, the class of random utility models with i.i.d. noise (IID-RUMs). We propose an algorithm for choice bandits, termed Winner Beats All (WBA), with a distribution dependent $O(\log T)$ regret bound under all these choice models. The challenge in our setting is that the decision space is $\Theta(n^k)$, which is large for even moderate $k$. Our algorithm addresses this challenge by extracting just $O(n^2)$ statistics from multiway choices and exploiting the existence of a unique 'best' arm to find arms that are competitive to this arm in order to construct sets with low regret. Since these statistics are extracted from the same choice observations, one needs a careful martingale analysis in order to show that these statistics are concentrated. We complement our upper bound result with a lower bound result, which shows that our upper bound is order-wise optimal. Our experiments demonstrate that for the special case of $k = 2$, our algorithm is competitive with previous dueling bandit algorithms, and for the more general case $k > 2$, outperforms the recently proposed MaxMinUCB algorithm designed for the MNL model.

## 1 Introduction

The dueling bandit problem has received a lot of interest in recent years [1, 2, 3, 4, 5, 6, 7, 8, 9, 10, 11, 12, 13, 14]. Here there are $n$ arms $\{1, \ldots, n\}$; on each trial $t$, the learner pulls a pair of arms $(i_t, j_t)$, and receives *relative feedback* indicating which of the two arms has a better quality/reward. In the regret minimization setting, the goal is to identify the 'best' arm(s) while also minimizing the regret due to playing sub-optimal arms in the learning (exploration) phase.

---

[*]Work done while at the University of Pennsylvania.

In many applications, however, it can be natural for the learner to pull more than two arms at a time, and seek relative feedback among them. For example, in recommender systems, it is natural to display several items or products to a user, and seek feedback on the most preferred item among those shown. In online advertising, it is natural to display several ads at a time, and observe which of them is clicked (preferred). In online ranker evaluation for information retrieval, one can easily imagine a generalization of the setting studied by Yue & Joachims [15], where one may want to "multi-leave" several rankers at a time to help identify the best ranking system while also presenting good/acceptable results to users using the system during the exploration phase. In general, there is also support in the marketing literature for showing customers more than two items at a time [16].

Motivated by such applications, we consider a framework that generalizes the dueling bandit problem to allow the learner to pull more than two arms at a time. Here, on each trial $t$, the learner pulls a set $S_t$ of up to $k$ arms (for fixed $k \in \{2, \ldots, n\}$), and receives relative feedback in the form of a multiway choice $y_t \in S_t$ indicating which arm in the set has the highest quality/reward. The goal of the learner is again to identify a 'best' arm (to be formalized below) while minimizing a suitable notion of regret that penalizes the learner for playing sub-optimal arms during the exploration phase. We term the resulting framework *choice bandits*.

In the (stochastic) dueling bandits framework, the underlying probabilistic model from which feedback is observed is a *pairwise comparison model*, which for each pair of arms $(i, j)$, defines a probability $P_{ij}$ that arm $i$ has higher reward/quality than arm $j$. In our choice bandits framework, the underlying probabilistic model is a *multiway choice model*, which for each set of arms $S \subseteq [n]$ with $|S| \le k$ and each arm $i \in S$, defines a probability $P_{i|S}$ that arm $i$ has the highest reward/quality in the set $S$.

We study choice bandits under a new class of choice models, that are characterized by the existence of a unique *generalized Condorcet winner* (GCW), which we define to be an arm that has larger probability of being chosen than any other arm in any choice set. This class includes as special cases the well-studied multinomial logit (MNL) [17, 18, 19] and multinomial probit (MNP) [20] choice models, and more generally, the class of random utility models with i.i.d. noise (IID-RUMs) [21, 22].

Our main contribution is a computationally efficient algorithm, termed *Winner Beats All* (WBA), that achieves a distribution dependent $O(n^2 \log n + n \log T)$ regret bound under any choice model that exhibits a unique GCW, where $T$ is the time-horizon. We complement our upper bound result with an order-wise lower bound of $\Omega(n \log T)$ for any *no-regret* algorithm, showing that our algorithm has asymptotically order optimal regret under our general class of choice models. If the underlying model is MNL, then WBA achieves an instance-wise asymptotically optimal regret bound, which is better than the regret bound for the recent MaxMinUCB algorithm under MNL [23].

The main challenge in designing an algorithm under our framework is that the space of exploration (number of possible sets the learner can play) is $\Theta(n^k)$ which is large even for moderate $k$. Therefore, it can be challenging to simultaneously *explore/learn* the choice sets with low regret out of the possible $\Theta(n^k)$ sets and *exploit* these low regret sets. We overcome these challenges by extracting just $O(n^2)$ pairwise statistics from the observed multiway choices under different sets, and using these statistics to find choice sets with low regret. Since these pairwise statistics are extracted from multiway choices under different sets, a technical challenge is to show that these statistics are concentrated. We resolve this challenge by using a novel coupling argument that couples the stochastic process generating choices with another stochastic process, and showing that pairwise estimates according to this other process are concentrated. We believe that our results for efficient learning under this large class of choice models that is considerably more general than the MNL class are of independent interest.

We also run experiments on several synthetic and real-world datasets. Our experiments on these datasets show that our algorithm for the special case of $k = 2$ is competitive as compared to previous dueling bandit algorithms, even though it is designed for a more general setting. For the case of $k > 2$, we compare our algorithm with the MaxMinUCB algorithm of [23] which was designed for the MNL model. We observe that our algorithm performs better in terms of regret than MaxMinUCB under all datasets (even under synthetic MNL datasets). We further observe that under several datasets the regret achieved by our algorithm for $k > 2$ is better than the regret for $k = 2$.

**Related Work.** There has been some recent interest in bandit settings where more than two arms are pulled at a time, although no work that we are aware of considers the types of general choice models that we do. (1) A related setting to ours is that of *multi-dueling bandits* [24, 25, 26], where the learner also pulls a set $S_t$ of $k$ items; however, the feedback received by the learner is assumed

to be drawn from a pairwise comparison model (in particular, the learner observes some subset of the $\binom{k}{2}$ possible pairwise comparisons among items in $S_t$). In contrast, in our choice bandits setting, the learner receives the outcome of a direct multiway choice among the items in $S_t$, generated from a multiway choice model. (2) In *combinatorial bandit with relative feedback* [23], the learner pulls a set $S_t$ of up to $k$ arms, and observes top-$m$ ordered feedback drawn according to the MNL model, for some $m \leq k$. In contrast, we only observe the (top-1) choice feedback from the set $S_t$ that is played, but, we study a much more general class of choice models than the MNL model. (3) Another related setting is that of *battling bandits* [27], where the learner pulls a multiset of $k$ arms and receives feedback indicating which arm was chosen. However, their setting considers a specific pairwise-subset (PS) choice model that is defined in terms of a pairwise comparison model, whereas we consider more general choice models. (4) In *stochastic click bandits* [28], the learner pulls an *ordered* set of $k$ arms/documents, and observes *clicks* on a subset of these documents, drawn according to an underlying *click model* which is a probabilistic model for click generation over an ordered set. However, click models in their setting are different than choice models in our setting, and neither can be cast as a special case of the other. (5) Another related setting is that of *best-of-k bandits* [29], where again the learner pulls a set $S_t$ of $k$ arms. Of the various types of feedback considered in [29], the *marked bandit* feedback corresponds to the type of feedback that we study, however, the choice models studied in [29] correspond only to a subclass of random utility choice models, and moreover, the analysis in [29] is in the PAC/pure exploration setting, while ours is in the regret minimization setting. (6) Other recent work has specifically considered active learning problems, either in the context of *dynamic assortment optimization* under MNL where the goal is to maximize expected revenue [30, 31, 32, 33, 34]; or in the context of *best arm(s) identification* under MNL or IID-RUMs [35, 36] in a PAC/pure exploration setting. (7) Finally, we also mention *combinatorial bandits*, which have a different goal but also involve pulling subsets of arms [37, 38, 39, 40]. See the supplementary material for more detailed discussion.

**Organization.** We set up the choice bandits problem in Section 2. We give our lower bound result in Section 3. We present our algorithm in Section 4, and its regret analysis in Section 5. We give experimental results in Section 6. We finally conclude with a brief discussion in Section 7. All the proofs can be found in the supplementary material.

## 2  Problem Setup and Preliminaries

In the choice bandits problem, there are $n$ arms $[n] := \{1, \ldots, n\}$, and a set size parameter $2 \leq k \leq n$. On each trial $t$, the learner pulls (selects/plays) a choice set $S_t \subseteq [n]$ of up to $k$ arms, i.e. with $|S_t| \leq k$, and receives as feedback $y_t \in S_t$, indicating the arm that is most preferred in $S_t$. We assume the feedback $y_t$ is generated probabilistically from an underlying *multiway choice model*, which defines for each $S \subseteq [n]$ such that $|S| \leq k$, and arm $i \in S$, a *choice probability* $P_{i|S}$ which corresponds to the probability that arm $i$ is the most preferred arm in $S$.[2] Before defining appropriate notions of 'best' arm and regret for the learner we will give some examples of multiway choice models.

**Random utility models with i.i.d. noise (IID-RUMs).** IID-RUMs are a well-known class of choice models that have origins in the econometrics and marketing literature [21, 41]. Under an IID-RUM, the (random) utility associated with arm $i \in [n]$ is given by $U_i = v_i + \epsilon_i$ where $v_i \in \mathbb{R}$ is a deterministic utility and $\epsilon_i \in \mathbb{R}$ is the noise drawn i.i.d. from a distribution $\mathcal{D}$ over reals. For a set $S$, the probability of choosing $i \in S$ is given by $P_{i|S} = \mathrm{Pr}\left(U_i > U_j, \forall j \in S \setminus \{i\}\right)$. We will sometimes also refer to $v_i$ as the weight of item $i$. Under any IID-RUM if $v_i > v_j$ for some $i, j \in [n]$ then arm $i$ will be more likely to be chosen than arm $j$ in any set. The IID-RUM class contains some popular models, such as the multinomial logit (MNL) [17, 18, 19], and multinomial probit (MNP) [20].

**Example 1** (MNL). *Under MNL, the noise distribution $\mathcal{D}$ is a Gumbel$(0, 1)$ and the probability $P_{i|S}$ of choosing an item $i$ from a set $S$ has the following closed form expression: $P_{i|S} := e^{v_i}/(\sum_{j \in S} e^{v_j})$. It is clear from this expression that arms with higher weights are more likely to be chosen.*

**Example 2** (MNP). *Under the MNP model, the noise distribution $\mathcal{D}$ is the standard Normal distribution $\mathcal{N}(0, 1)$, however, unlike the MNL there is no closed form expression for the choice probabilities.*

Under IID-RUMs there is a clear notion of 'best' arm: an arm that has the highest weight $\max_{i \in [n]} v_i$. We now define a strictly more general class of models where there is a clear notion of 'best' arm.

**A New Class of Choice Models.** We introduce a new class of multiway choice models that are characterized by the following condition that requires the existence of a unique 'best' arm.

**Definition 1** (Generalized Condorcet Condition (GCC)). *A choice model is said to satisfy the GCC condition if there exists a unique arm $i^* \in [n]$ such that for every choice set $S \subseteq [n]$ that contains $i^*$, we have $P_{i^*|S} > P_{j|S}$ for all $j \in S \setminus \{i^*\}$.*

Intuitively, the above condition requires the existence of a unique arm that is always (stochastically) preferred to all other arms, no matter what other arms are shown with it. This condition is a generalization of the Condorcet condition studied for pairwise comparison models [6, 11]. Just as the Condorcet condition need not be satisfied for all pairwise comparison models, similarly, GCC need not be satisfied by all multiway choice models. Below we show that the GCC condition is satisfied for all IID-RUMs subject to a minor technical condition.

**Lemma 1** (IID-RUMs satisfy GCC). *For any IID-RUM choice model with utility for arm $i \in [n]$ given by $U_i = v_i + \epsilon_i$, the GCC condition is satisfied if $|\operatorname{argmax}_{i \in [n]} v_i| = 1$.*

In this paper, we study the class of all choice models where the GCC is satisfied. Under GCC, we will refer to this unique 'best' arm as the generalized Condorcet winner (GCW) and denote it by $i^*$. Note that for any set $S$ containing the GCW $i^*$, we must have $P_{i^*|S} \geq \frac{1}{|S|}$.

**Regret Notion.** Similar to dueling bandits, the goal of the learner in our setting is to identify the best arm while also playing good/competitive sets with respect to this arm during the exploration phase.[3] Hence, our notion of regret measures the sub-optimality of a choice set $S$ relative to $i^*$, and is a generalization of the regret defined by [23] for the special case of MNL choice model. Moreover, under our notion of regret it is optimal to play $S^* = \{i^*\}$, i.e. regret of playing $S^*$ is 0. The regret of a set is defined to be the sum of regret due to individual arms in the set, and the regret for an arm corresponds to the 'margin' by which the best arm $i^*$ beats this arm. In other words, the regret of an arm corresponds to the *shortfall in preference probability* due to pulling this arm over the 'best' arm.

**Definition 2.** *The regret $r(S)$ for $S \subseteq [n]$ is defined as: $r(S) := \sum_{i \in S} \left( P_{i^*|S \cup \{i^*\}} - P_{i|S \cup \{i^*\}} \right)$.*

This notion of regret can be interpreted as: $r(S)$ is the sum over all arms $i \in S$, the fraction of consumers that will choose $i^*$ minus the fraction of consumers that will choose $i$ when $i^*$ is played together with $S$. It is easy to see that $r(\{i^*\}) = 0$, and $0 \leq r(S) \leq |S|$ for any set $S \subseteq [n]$.

**Example 3.** *Consider a choice model where arm $1$ is the GCW, and for each set $S$ containing arm $1$, we have $P_{1|S} = 0.51$ and $P_{i|S} = \frac{0.49}{|S|-1} \, \forall i \in S \setminus \{1\}$. Then $r(\{1, \ldots, m\}) = 0.51 \times (m-1) - 0.49$.*

In the above example, the regret increases linearly as we increase $m$. The following gives an example where the arms are much more 'competitive' and regret is smaller.

**Example 4.** *Consider the MNL choice model with weights $v_1 = \log(1 + \epsilon)$, for $\epsilon > 0$, and $v_2 = \cdots = v_n = 0$. Then $r(\{1, \cdots, m\}) = \sum_{i \in S} \frac{e^{v_1} - e^{v_i}}{\sum_{j \in S} e^{v_j}} = \frac{\epsilon(m-1)}{m+\epsilon}$.*

The regret here increases much more slowly in terms of $m$. Note that our regret is not necessarily well-defined in the dueling bandits setting, due to the need to consider choice probabilities for sets of size 3 even when one plays only sets of size 2. In the supplementary material, we give results for an additional notion of regret that is a direct generalization of the dueling bandit regret, and allows for a more direct comparison between our framework and the dueling bandits framework.

Under the above notion of regret, the goal of an algorithm $\mathcal{A}$ is to minimize its cumulative regret over $T$ trials defined as: $R(T) = \sum_{t=1}^{T} r(S_t)$.

## 3 A Fundamental Lower Bound

In this section we present a regret lower bound for our choice bandits problem. We say that an algorithm is *strongly consistent* under GCC if its expected regret over $T$ trials is $o(T^a)$ for any $a > 0$ under any model in this class. Before presenting the lower bound let us define the following distribution dependent quantities.

$$\Delta_{i^*i|S} = \frac{P_{i^*|S} - P_{i|S}}{P_{i^*|S} + P_{i|S}}, \quad \Delta_{\max}^{\text{GCC}} := \max_{S:|S| \leq k} \max_{i \in S} \Delta_{i^*i|S}, \quad \Delta_{\min}^{\text{GCC}} := \min_{S:|S| \leq k} \min_{i \in S} \Delta_{i^*i|S}. \quad (3.1)$$

The following theorem presents a lower bound for any strongly consistent algorithm.

**Theorem 1.** *Given a set of arms $[n]$, choice set size bound $k \leq n$, parameter $\Delta \in (0, 1)$, and any strongly consistent algorithm $\mathcal{A}$ under GCC, there exists a GCC choice model with $\Delta_{\min}^{\text{GCC}} = \Delta$ such that when choice outcomes are drawn from this model we have*

$$\liminf_{T \to \infty} \frac{\mathbf{E}\left[R(T)\right]}{\log T} = \Omega\left(\frac{n-1}{\Delta}\right),$$

*where $T$ is the time-horizon. If the underlying model is MNL with parameters $v_1, v_2, \cdots v_n \in \mathbb{R}$, then:* $\liminf_{T \to \infty} \frac{\mathbf{E}[R(T)]}{\log T} = \Omega\left(\sum_{i \in [n] \setminus \{i^*\}} \frac{1}{\Delta_{i^*i}^{\text{MNL}}}\right)$ *where* $\Delta_{i^*i}^{\text{MNL}} = \frac{e^{v_{i^*}} - e^{v_i}}{e^{v_{i^*}} + e^{v_i}}$, *for $i \in [n] \setminus \{i^*\}$.*

**Discussion.** The above bound shows that any algorithm for the choice bandits problem needs to incur $\Omega(n \log T)$ regret in the worst case. Note that the above lower bound does not depend on the choice set size parameter $k$. If the choices are generated from an underlying MNL model, then the above theorem gives an instance-dependent lower bound for the regret of any algorithm. Note that [23] also provided a lower bound under MNL for our notion of regret, however, their bound depends on the worst-case gap between $i^*$ and any other arm $i \neq i^*$, while we provide a more fine-grained bound under MNL which depends on gaps between $i^*$ and each individual arm $i \in [n]$.

In order to prove the above bound we construct a pair of instances that have different GCW arms, and use the information divergence lemma of [42] in order to characterize the minimum number of samples needed in order to collect the 'information' needed to separate these two instances.

**Remark 1.** *In order to prove a lower bound for our choice bandits problem one may also be able to use the lower bound given in [43], by casting our problem as a structured bandit problem. However, the lower bound of [43] is in terms of the solution of a linear program, and one will then need to design a distribution over hard instances in order to quantify the solution of this linear program in terms of the gap parameter $\Delta_{\min}^{\text{GCC}}$. One of the main novelty of our bound is this construction of hard instances that allows us to quantify the lower bound in terms of $\Delta_{\min}^{\text{GCC}}$.*

## 4 Algorithm

In this section we will present our algorithm for the choice bandits problem, termed Winner Beats All (WBA). WBA divides its execution into rounds and each round can contain multiple trials. We will use $r$ as an global index for a round, and $t$ as an global index for a trial. For each round $r$, WBA maintains a set $A_r$ of active arms, which are a set of arms for which the algorithm is still not confident enough that these are 'bad' arms. Note that an arm that is inactive in a particular round, can become active in a later round. We also maintain a set $Q$ that is initialized to being empty at the beginning of each round and keeps track of the arms in $A_r$ that have been played so far in the round.

Given a trial $t$ belonging to round $r$, WBA selects a special arm termed the 'anchor' arm, and a set $S \subseteq A_r \setminus Q$ (arbitrarily) of up to $k - 1$ arms in $A_r$ that have not been played so far in round $r$. The set $S$ and $a_t$ are selected such that $a_t$ empirically performs better than each arm in $S$. The set $S$ is then played together with arm $a_t$ (if $|S| < k - 1$, then other arbitrary arms from $A_r$ are added to the played set). The anchor arm is updated in every trial and is chosen so that one can *quickly* find evidence that arms in $S$ are not good.

Let $y_t$ be the feedback received in trial $t$ when $S_t$ was played including anchor $a_t$. For all $i, j \in [n]$, let $N_{ij}(t)$ denote the number of times (up to round $t$) that either arm $i$ or $j$ was chosen when arm $j$ is the anchor, i.e. $N_{ij}(t) := \sum_{t'=1}^{t} \mathbb{1}(a_{t'} = j, \{i, j\} \subseteq S_{t'}, y_{t'} \in \{i, j\})$.

For each $i, j \in [n]$ and trial $t$, such that $N_{ij}(t) > 0$, the algorithm maintains an estimate of the *marginal probability* of arm $i$ beating the arm $j$ as

$$\hat{P}_{ij}(t) := \frac{1}{N_{ij}(t)} \sum_{t'=1}^{t} \mathbb{1}(a_{t'} = j, \{i, j\} \subseteq S_{t'}, y_{t'} = i), \tag{4.1}$$

which is the fraction of times $i$ was selected (compared to $j$) when both $i$ and $j$ were played together and $j$ was the anchor. (When $N_{ij}(t) = 0$, we can simply take $\hat{P}_{ia}(t)$ to be $1/2$.) Similar to [44], let us define an *empirical divergence* $I_i(t, S)$ which provides a certificate that an arm $i$ is worse than (some) arms in $S$, as $I_i(t, S) = \sum_{j \in S} \mathbb{1}[\hat{P}_{ij}(t) \leq \frac{1}{2}] \cdot N_{ij}(t) \cdot d(\hat{P}_{ij}(t), \frac{1}{2})$, where $d(\hat{P}_{ij}, \frac{1}{2})$ is the KL-divergence defined as $d(P, Q) = P \log(\frac{P}{Q}) + (1 - P) \log(\frac{1-P}{1-Q})$, for $P, Q \in [0, 1]$. If $I_i(t, S)$ is

0, it means that arm $i$ is empirically at least as good as all other arms in $S$, and a higher $I_i(t, S)$ would suggest that arm $i$ is most likely 'bad'. For a constant $C$, we define the condition $\mathcal{J}_i(t, C)$ for arm $i \in [n]$ and round $t$ as $\mathcal{J}_i(t, C) = \mathbb{1}\left\{\exists S \subseteq [n] : I_i(t, S) \geq |S| \log(nC) + \log(t)\right\}$. If $\mathcal{J}_i(t, C) = 1$ for some $i$, it means that there exists a certificate $S$ to show that $i$ is not likely the best arm as it loses to some arms in $S$ by a large 'margin'.[4] The larger the set $S$ the larger the margin needs to be. This condition can be evaluated in polynomial time by computing $\operatorname{argmax}_{S \subseteq [n]} I_i(t, S) - |S| \cdot \log(nC)$ and checking if it is greater than $\log(t)$ (details in supplementary material).

Finally, let $t$ be the final round in a round $r$. In order to decide which arms should be included in the next set of active arms $A_{r+1}$ we simply check the condition $\mathcal{J}_i(t, C)$ for each $i \in [n]$ and include all arms for which $\mathcal{J}_i(t, C) = 0$ holds. Note that the set of active arms $A_{r+1}$ can be empty, in which case we will simply play the anchor arm until it becomes non-empty in the future. The anchor arm in each trial is the arm which empirically beats the maximum number of unplayed arms in the current round. Detailed pseudo-code for WBA is given in Algorithm 1.

## 5 Regret Analysis

In this section we will prove a regret upper bound for our WBA algorithm. The following theorem gives the upper bound.

**Theorem 2.** *Let $n$ be the number of arms, $k \leq n$ be the choice set size parameter, and $i^*$ be the GCW arm . If the multiway choices are drawn according to a GCC choice model with $\Delta_{\min}^{\mathrm{GCC}}$ and $\Delta_{\max}^{\mathrm{GCC}}$ defined in Equation 3.1, then for any $C \geq 1/(\Delta_{\min}^{\mathrm{GCC}})^4$, the expected regret incurred by WBA is upper bounded by*

$$\mathbf{E}\left[R(T)\right] \leq O\left(\frac{n^2 \log n}{(\Delta_{\min}^{\mathrm{GCC}})^2}\right) + O\left(n \log(TC) \cdot \frac{\Delta_{\max}^{\mathrm{GCC}}}{(\Delta_{\min}^{\mathrm{GCC}})^2}\right),$$

*where $T$ is the (unknown) time-horizon. If the underlying model is MNL with weights $v_1, \cdots, v_n \in \mathbb{R}$, then for any $C \geq 1/(\Delta_{\min}^{\mathrm{MNL}})^4$, we have*

$$\mathbf{E}\left[R(T)\right] \leq O\left(\frac{n^2 \log n}{(\Delta_{\min}^{\mathrm{MNL}})^2}\right) + O\left(\sum_{i \in [n] \setminus i^*} \frac{\log(TC)}{\Delta_{i^* i}^{\mathrm{MNL}}}\right),$$

*where $\Delta_{i^* i}^{\mathrm{MNL}} = \frac{e^{v_{i^*}} - e^{v_i}}{e^{v_{i^*}} + e^{v_i}}$ and $\Delta_{\min}^{\mathrm{MNL}} := \min_{i \neq i^*} \Delta_{i^* i}^{\mathrm{MNL}}$.*

**Remark 2** (Selecting $C$). *A value of $T^4$ for the parameter $C$ suffices for Theorem 2 to hold, giving a regret upper bound of $O(\log(TC)) = O(\log(T^5)) = O(\log(T))$. (If $T$ is not known, one can use the doubling trick.) To see this note that in order to obtain any non-trivial upper bound for our algorithm, $\Delta_{\min}$ has to be larger than $1/T$. Hence, either $\Delta_{\min}$ is upper bounded by $1/T$, or the instance is too hard to allow any non-trivial upper bound. Therefore, $C \geq T^4$ would suffice whenever the instance is not already too hard. We actually believe setting $C = T^4$ may be somewhat pessimistic (it arises from taking a union bound over all possible states of the algorithm in our regret analysis – indeed, in our experiments, we set $C = 1$ for all datasets, and our algorithm still demonstrates sublinear regret with this choice – but it certainly suffices, and the regret bound with $C = T^4$ is at most a constant factor 5 times what one might get with $C = 1$ if the regret bound holds in that case.*

**Discussion.** The above theorem yields a $O(n^2 \log n + n \log T)$ upper bound on regret. Comparing this bound with the lower bound given in Section 3, one can observe this upper bound is *asymptotically* order-optimal. This upper bound is similar (in order-wise sense) to the upper bounds obtained for some popular dueling bandit algorithms such as RUCB [6], RMED [44], DTS [45] etc. It is also important to note that our regret bound does not depend directly on the choice set size $k$. However, the behavior of this bound is more subtle and depends on the specific multiway choice model through the gap parameters $\Delta_{\max}^{\mathrm{GCC}}$ and $\Delta_{\min}^{\mathrm{GCC}}$. We also note that while in general the regret can behave differently for different models, in our experiments, we find that there are choice models (including some in real data) where our algorithm empirically achieves smaller regret when allowed to play sets of size $k > 2$ as compared to $k = 2$. If the underlying model is MNL, then our algorithm achieves asymptotically

| **Algorithm 1** Winner Beats All (WBA) |
| --- |

1: **Input**: set of arms $[n]$, size of choice set $k$, parameter $C$
2: $t \leftarrow 1, r \leftarrow 1, A_r \leftarrow [n], a_t \leftarrow \text{Unif}([n]), Q \leftarrow \emptyset$
3: $\hat{P}_{ij} \leftarrow \frac{1}{2}, \forall i, j \in [n]$
4: **while** $t \leq T$ **do**
5:     Select largest $S \subseteq A_r \setminus \{Q \cup a_t\}$ with $|S| \leq k - 1$ and $\hat{P}_{ia_t} \leq \frac{1}{2}, \forall i \in S$
6:     Let $S_t \leftarrow S \cup \{a_t\}$; while $|S_t| < k$ and $A_r \setminus S_t \neq \emptyset$: add an (arbitrary) arm from $A_r \setminus S_t$ to $S_t$
7:     Play set $S_t$ and receive $y_t \in S_t$ as feedback; $Q \leftarrow Q \cup S$
8:     For all $i \in S_t$, calculate $\hat{P}_{ia_t}(t)$ and $\mathcal{J}_i(t, C)$
9:     **if** $\nexists i \in A_r \setminus \{Q \cup a_t\}$ such that $\hat{P}_{ia_t}(t) \leq \frac{1}{2}$ **then**
10:         $a_{t+1} \leftarrow \text{argmax}_{i \in [n]} \sum_{j \in [n] \setminus Q} \mathbb{1}[\hat{P}_{ji}(t) \leq \frac{1}{2}]$
11:     **else**
12:         $a_{t+1} \leftarrow a_t$
13:     **if** $Q = A_r$ or $S = \emptyset$ **then**
14:         $A_{r+1} \leftarrow \emptyset, r \leftarrow r + 1$
15:         **for** $i \in [n]$ **do**
16:             **if** $\mathcal{J}_i(t, C) = 0$, **then** $A_r \leftarrow A_r \cup \{i\}$
17:         $a_{t+1} \leftarrow \text{argmax}_{i \in [n]} \sum_{j \in [n]} \mathbb{1}[\hat{P}_{ji}(t) \leq \frac{1}{2}], Q \leftarrow \emptyset$
18:     $t \leftarrow t + 1$

Figure 1: Datasets used in our experiments

1. **MNL-Exp:** MNL weights drawn i.i.d. from $\text{Exp}(\lambda = 3.5)$;

2. **MNL-Geom:** Geometrically decreasing MNL weights: $1, \frac{1}{2}, \ldots, \frac{1}{2^{n-1}}$;

3. **GCC-One:** GCC model defined in Example 3;

4. **GCC-Two:** GCC model similar to Example 3, but with different choice probabilities;

5. **GCC-Three:** GCC model similar to Example 3, but with different choice probabilities;

6. **Sushi:** Choice model extracted from the Sushi dataset [47];

7. **Irish-Dublin:** Choice model extracted from Irish-Dublin election dataset;

8. **Irish-Meath:** Choice model extracted from Irish-Meath election dataset.

optimal instance-wise regret that does not depend on $k$. This instance-wise bound under MNL is an improvement over the upper bound for the MaxMinUCB algorithm under MNL for (top-1) choice feedback which depends on worst-case gap parameters [23]. An important point to note is that we do not need a specialized algorithm for MNL in order to achieve an instance-wise bound under MNL.

**Proof Ideas.** Our algorithm is based on the idea of isolating a 'good' anchor arm and playing arms that are competitive against this anchor. Hence, in order to prove a regret upper bound we need to show that the GCW $i^*$ would eventually beat every other arm $i$, i.e. $\hat{P}_{i^*i}(t)$ (Equation 4.1) would eventually become larger than $1/2$. In this case $i^*$ would become the anchor arm. However, an important technical challenge here is to bound the deviation in these pairwise estimates $\hat{P}_{i^*i}(t)$ obtained from multiway choices. In the past, [46] have shown that if one uses rank breaking to extract pairwise estimates under the MNL model, then these pairwise estimates will be concentrated. However, this concentration result relies crucially on the independence from irrelevant attributes (IIA) property of MNL which states that for any two arms, the odds of choosing one over the other in any set remains the same *regardless of which set is shown*. This concentration result does not apply to our setting as the IIA property does not hold for general GCC models beyond the MNL.

Below we outline a novel coupling argument that allows us to prove concentration for the extracted pairwise estimates between the GCW arm $i^*$ and any other arm $i \in [n]$

**Lemma 2** (Concentration). *Consider a GCC choice model with GCW $i^*$. Fix $i \in [n]$. Let $S_1, \cdots, S_T$ be a sequence of subsets of $[n]$ and $y_1, \cdots, y_T$ be a sequence of choices according to this model, let $\mathcal{F}_t = \{S_1, y_1, \cdots, S_t, y_t\}$ be a filtration such that $S_{t+1}$ is a measurable function of $\mathcal{F}_t$. We have*

$$\Pr(\hat{P}_{i^*i}(t) \leq P_{i^*i}^{\text{GCC}} - \epsilon \text{ and } N_{i^*i}(t) \geq m) \leq e^{-d(P_{i^*i}^{\text{GCC}} - \epsilon, P_{i^*i}^{\text{GCC}}) \cdot m} \quad (5.1)$$

*where $P_{i^*i}^{\text{GCC}} = \min_{S:|S| \leq k, \{i^*, i\} \subseteq S} \frac{P_{i^*|S}}{P_{i^*|S} + P_{i|S}}$, and $d(\cdot, \cdot)$ is the KL-divergence.*

*Proof Sketch.* Let us consider an alternate process for generating multiway choices $y_t'$ from sets $S_t$. In this process, given any $t$ and a set $S_t$ such that $i^*, i \in S_t$ with $a_t = i$, we first generate a Bernoulli random variable $X_t$ with probability $P_{i^*|S} + P_{i|S}$. If $X_t = 0$ we set $y_t' = j$ with probability $\frac{P_{j|S}}{1 - P_{i^*|S} - P_{i|S}}$, for $j \in S \setminus \{i, i^*\}$. If $X_t = 1$ then we sample another Bernoulli random variable $Z_t$ with probability $P_{i^*i}^{\text{GCC}}$. If $Z_t = 1$ then we let $y_t' = i^*$, otherwise if $Z_t = 0$ we set $y_t' = i$. Let $P_{i^*i|S_t} = P_{i^*|S_t}/(P_{i^*|S_t} + P_{i|S_t})$. Now, we couple $y_t'$ and $y_t$ as follows: if $y_t' \in S_t \setminus \{i\}$ then we let $y_t = y_t'$, otherwise if $y_t' = i$ then we let $y_t = i^*$ with probability $(P_{i^*i|S_t} - P_{i^*i}^{\text{GCC}})/(1 - P_{i^*i}^{\text{GCC}})$ and let $y_t = i$ with probability $(1 - P_{i^*i|S_t})/(1 - P_{i^*i}^{\text{GCC}})$. One can verify that $y_t$ is distributed according to the correct underlying choice distribution. It is now easy to observe that the estimates $\hat{P}_{i^*i}(t)$ under $y_t$ will always be larger than the estimates $\hat{P}'_{i^*i}(t)$ under $y_t'$, hence, we will have that

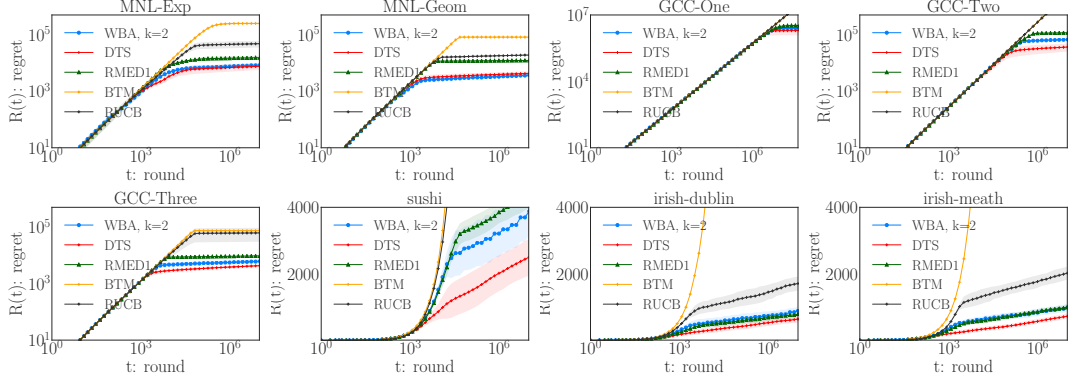

Figure 2: Regret v/s trials for our algorithm WBA (for $k = 2$) against dueling bandit algorithms (DTS, BTM, RUCB and RMED1) (the shaded region corresponds to std. deviation). As can be observed, our algorithm is competitive against these algorithms.

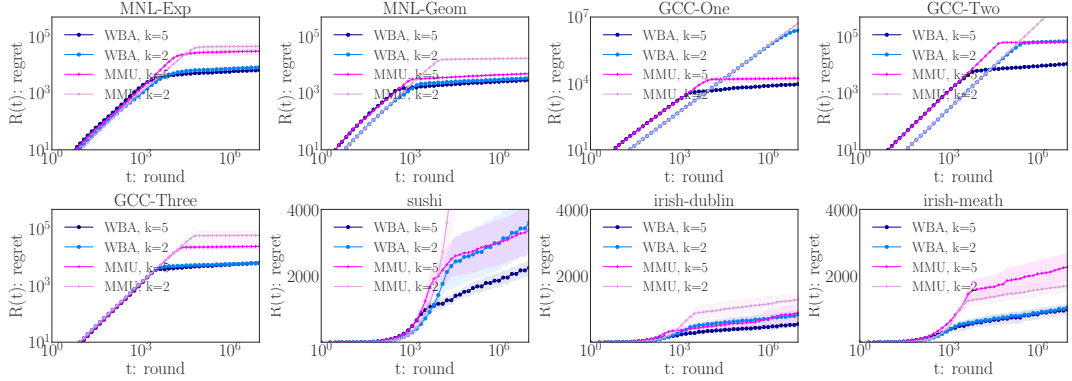

Figure 3: Regret v/s trials for our algorithm WBA against the MaxMinUCB (MMU) algorithm for $k = 2$ and $k = 5$ (the shaded region corresponds to std. deviation). We observe that our algorithm is better than MaxMinUCB on all datasets for both values of $k$. We further observe that under several datasets the regret achieved by our algorithm for $k > 2$ is better than the regret of our algorithm for $k = 2$.

$\Pr(\hat{P}_{i^*i}(t) \le x) \le \Pr(\hat{P}'_{i^*i}(t) \le x)$ for any $x > 0$. One can then show concentration for the coupled estimates $\hat{P}'_{i^*i}(t)$, and use it to bound the deviation in $\hat{P}_{i^*i}(t)$. $\qquad\square$

Note that the above lemma only shows concentration for the pairwise estimates $\hat{P}_{i^*i}(t)$ between $i^*$ and any other arm $i \in [n]$, but not for estimates $\hat{P}_{ij}(t)$ between two arbitrary arms $i \in [n]$ and $j \in [n]$. However, in order to prove our result we only need concentration of estimates between $i^*$ and any other arm $i \in [n]$. We believe that the above concentration lemma is of independent interest, and might be useful in other learning from multiway choice settings beyond MNL.

Once we have bounded the deviation for the pairwise estimates, we bound the number of rounds $r$ in which $i^*$ is not a part of the active set $A_r$. We then bound the expected number of times that there exists an arm $i$ such that $\hat{P}_{i^*i}(t) < \frac{1}{2}$, thus bounding the number of trials until $i^*$ becomes the anchor. Finally, once $i^*$ is the anchor arm, we bound the regret incurred due to sub-optimal arms.

## 6    Experiments

We compared the performance of our WBA algorithm and other existing algorithms on our choice bandit problem under different choice models. The first two choice models were MNL models, the next three were from the GCC class, and the last three we extracted from real-world datasets. Details of these models are in Figure 1 (additional details can be found in the supplementary material).

Below we describe the different sets of experiments that were performed. Each experiment was repeated 10 times. The value of $n$ was 100 for all synthetic datasets, 16 for Sushi, 8 for Irish-Dublin, and 12 for Irish-Meath. The parameter $C$ in our algorithm was set to 1.

**Comparison with Dueling Bandit Algorithms** ($k = 2$)**.** For the special case of $k = 2$, we compared our algorithms with a representative set of dueling bandit algorithms (RMED1 [11], DTS [45], RUCB [6], BTM [2]). Note that the purpose of these experiments is merely to perform a sanity check and ensure that our algorithm performs reasonably well compared with dueling bandit baselines when $k = 2$; the goal is not to argue that our choice bandit algorithm beats the state-of-the-art for the specialized dueling bandit ($k = 2$) setting. We set $\alpha = 0.51$ for RUCB and DTS, and $f(K) = 0.3K^{1.01}$ for RMED, and $\gamma = 1.3$ for BTM. Figure 2 contain plots for these comparisons. Our algorithm either performs better or similar to RMED1, RUCB, and BTM on all datasets; and is competitive with DTS on most of the datasets.

**Comparison with MaxMinUCB Algorithm [23]** ($k > 2$)**.** We compared the performance of our algorithm with the recent MaxMinUCB algorithm [23] that was designed and analyzed primarily for MNL choice models under the same notion of regret as ours. [5] We set the parameter $\alpha$ to be $0.51$ for MaxMinUCB. Figure 3 contain plots for these experiments for $k = 2$ and $k = 5$. We observe that our algorithm is much better in terms of regret than MaxMinUCB under all datasets for both values of $k$. One should note that WBA performs better than MaxMinUCB even under the MNL datasets, even though MaxMinUCB is specialized to MNL while our algorithm works under more general models. We further observe that under several datasets (GCC-One, GCC-Two, Sushi, Irish-Dublin) the regret achieved by our algorithm for $k > 2$ is better than for $k = 2$. Note that even though our study of more general choice feedback is motivated by applications where it might be desirable to pull sets of size larger than 2 due to reasons other than improving regret, these experimental results show that there exist settings of choice models (including some in real data) where our algorithm empirically achieves a smaller regret when allowed to play sets of size $k > 2$ as compared to $k = 2$.

## 7 Conclusion

We have introduced a new framework for bandit learning from choice feedback that generalizes the dueling bandit framework. Our main result is to show that computationally efficient learning is possible in this more general framework under a wide class of choice models that is considerably more general than the previously studied class of MNL models. Our algorithm for this general setting, termed Winner Beats All (WBA), achieves order-wise optimal regret for the general class of GCC models. For the special case $k = 2$, WBA is competitive with previous dueling bandit algorithms; for $k > 2$, WBA outperforms the recently proposed MaxMinUCB (MMU) algorithm even on MNL models for which MMU was designed.

## Acknowledgement

Thanks to Aadirupa Saha for early discussions related to this work, and to the anonymous referees for helpful comments. This material is based upon work supported in part by the US National Science Foundation (NSF) under Grant Nos. 1717290 and 1934876. Any opinions, findings, and conclusions or recommendations expressed in this material are those of the authors and do not necessarily reflect the views of the National Science Foundation.

## Broader Impact

The purpose of this paper is to understand whether efficient learning is possible in a bandit setting where one does not receive quantitative feedback for an individual arm but rather relative feedback in the form of a multiway choice. It is well-known that quantitative judgments of humans can have biases; our algorithm, which learns from relative multiway choices, can help alleviate these biases. Moreover, by receiving larger choice sets from our algorithm, humans can have a better sense of the quality distribution of arms, and can make more informed choices.

Another advantage of our setting is that we do not rely on historic data as our data collection is online. Hence, one does not need to worry about past biases being reflected in the choice datasets. However, one has to be cautious about the use of our algorithm in applications where arms represent individuals/entities such as job applicants, property renters etc. In these applications, the choices of people can be biased against certain individuals/groups, thereby hurting the chances of these individuals/groups to be selected by our algorithm. Here, depending on the application, one might need to consider imposing some form of fairness constraints on the choice sets output by our algorithm in order to prevent any discrimination against such individuals/groups.

## Footnotes

[2]Note that for the special case of $k = 2$, our framework reduces to dueling bandits; the pairwise comparison probabilities $P_{ij} := \mathrm{Pr}\,(i \succ j)$ in dueling bandits can be viewed as pairwise choice probabilities $P_{i|\{i,j\}}$.

[3]Note that we are *not* working in the pure exploration setting, where all sets incur equal cost during exploration.

[4]Note that the above condition is similar to condition used in [44], except that they only use the set $[n]$ as a certificate instead of all possible subsets $S \subseteq [n]$. In our analysis and experiments will show that this condition is an improvement over the condition used in [44] for the case of dueling bandits.

[5] We also considered the SelfSparring algorithm of [26] and the battling bandit algorithms of [27], which are applicable to choice models defined in terms of an underlying pairwise comparison model $P$. However, these algorithms all return *multisets* $S_t$, and any simple reduction of such multisets to strict sets as considered in our setting (as well as the setting of [23]) can end up throwing away important information learned by the algorithms, resulting in a comparison that could be unfair to those algorithms. We did explore such reductions and our algorithm easily outperformed them, but we chose not to include the results here due to this issue of fairness. (Moreover, under the MNL model, [23] already established that MaxMinUCB outperforms those algorithms – presumably under similar reductions – so in the end, we decided such a comparison would provide little additional value here.)

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
