[Supplementary Material]

# Choice Bandits
## Supplementary Material

## A  Organization

We provide additional discussion about the related work in Appendix B. We provide the proof of our regret lower bound (Theorem 1) in Appendix C. We prove a concentration inequality for pairwise estimates in Appendix D. We then provide the proof of our regret upper bound (Theorem 2) in Appendix E. In Appendix F we provide additional details about our experimental setup. In Appendix G we provide experimental results for an alternate notion of regret. Appendix H contains some technical lemmas used in the proof of the upper bound result in Theorem 2.

## B  Related Work

There has been some recent interest in bandit settings where more than two arms are played at once (although no previous work considers choice models at the level of generality we do). We review related work here and provide a summary in Table 1.

**Multi-dueling bandits:** In *multi-dueling bandits* [1, 2, 3], the learner pulls a set $S_t$ of $k$ items; however, the feedback received by the learner is assumed to be drawn from a pairwise comparison model (in particular, the learner observes some subset of the $\binom{k}{2}$ possible pairwise comparisons among items in $S_t$). In contrast, in our choice bandits setting, the learner receives the outcome of a direct multiway choice among the items in $S_t$, generated from a multiway choice model.

**Combinatorial bandits:** In *combinatorial (semi) bandits* [4, 5, 6, 7], each arm $i$ is associated with an unknown random variable (stochastic reward) $Y_i$; the learner pulls a set $S_t$ of up to $k$ arms, and observes the realized rewards $y_t(i)$ for all arms $i$ in $S_t$. In contrast, we only observe the arm that is chosen from the set $S_t$ that is played.

**Combinatorial bandits with relative feedback:** In this very recent framework [8], the learner pulls a set $S_t$ of up to $k$ arms, and observes top-$m$ ordered feedback drawn according to the MNL model, for some $m \leq k$. In contrast, we only observe the (top-1) choice feedback from the set $S_t$ that is played. Moreover, we study a much more general class of choice models than the MNL model studied by them.

**Stochastic Click Bandits:** In *stochastic click bandits* [9], the learner pulls an *ordered* set of $k$ arms/documents, and observes *clicks* on a subset of these documents, drawn according to an underlying *click model* which is a probabilistic model for click generation over an ordered set. However, click models in their setting are different than choice models in our setting, and neither can be cast as a special case of the other.

**Battling Bandits:** Another related setting is that of *battling bandits* [10], where the learner pulls a set $S_t$ of *exactly* $k$ arms and receives a feedback indicating which arm was chosen. However, their setting considers a specific pairwise-subset (PS) choice model that is defined in terms of a pairwise comparison model, whereas we consider much more general choice models.

**Preselection Bandits:** There has been a recent framework called *preselection bandits* [11] where two settings are considered: (1) where the learner pulls a set $S_t$ of size *exactly* $k$, (2) where the learner pulls a set $S_t$ of any size less than $n$. In both settings the learner receives feedback drawn from the MNL model. Firstly, the two settings considered by this paper are different than our setting where the learner plays a set of size up to $k$. Secondly, we study a much

| Problem | Rep. Paper | Arms Pulled in Round $t$ | Feedback in Round $t$ | Goal |
|---|---|---|---|---|
| Dueling Bandits | [19] | $(i_t, j_t) \in [n]^2$ | $y_t \in \{i_t, j_t\}$ | Min. regret w.r.t. best arm |
| Multi-dueling Bandits | [3] | $S_t \in [n]^k$ | $Y_t = \{0, 1, \emptyset\}^{k \times k}$ | Min. regret w.r.t. best arm |
| Combinatorial Bandits | [6] | $S_t \in \mathcal{S} \subseteq 2^{[n]} : \|S_t\| \le k$ | $y_t(i) \in \mathbb{R} \; \forall i \in S_t$ | Min. regret w.r.t. top-$k$ arms |
| Com. Ban. Relative Feed. | [8] | $S_t \subseteq [n] : \|S_t\| \le k$ | $O_t \subseteq S_t, \|O_t\| \le m$ | Min. reg. w.r.t. best arm (MNL) |
| Battling Bandits | [10] | $S_t \in [n]^k$ | $y_t \in S_t$ | Min. reg. w.r.t. best arm (PS) |
| Stochastic Click Bandits | [9] | $O_t \subseteq [n] : \|O_t\| = k,$ | $y_t \subseteq O_t$ | Max. expected clicks |
| Dynamic Assortment | [13] | $\{0\} \cup S_t \subseteq [n] : \|S_t\| \le k$ | $y_t \in S_t$ | Max. expected revenue |
| Choice Bandits | This paper | $S_t \subseteq [n] : \|S_t\| \le k$ | $y_t \in S_t$ | Min. regret w.r.t. best arm |

Table 1: Overview of related work in regret minimization settings. There are several definitions of 'best' arm; the reader is encouraged to refer to the relevant papers and to our problem setting for details. (Note: in multi-dueling bandits, $\emptyset$ denotes no feedback; in stochastic click bandits, $O_t$ denotes an ordered set; in combinatorial bandits, $\mathcal{S}$ denotes a set of allowed subsets; in dynamic assortment optimization, $0$ denotes the "no-purchase" option.)

more general class of choice models than the MNL model studied by them.

**Dynamic assortment optimization:** In *dynamic assortment optimization* [12, 13, 14, 15, 16], there are $n$ products and each product is associated with a revenue. The learner plays an assortment $S_t$ of up to $k$ products, and observes a feedback indicating which (if any) of the products was purchased; the goal of the learner is to maximize the expected revenue.

**Best-of-$k$ bandits (PAC setting).** [17] consider a *best-of-$k$ bandits* setting, where again the learner pulls a set $S_t$ of $k$ arms; however here each arm $i$ is associated with an unknown random variable (stochastic reward) $Y_i$. Of the various types of feedback that are considered, the *marked bandit* feedback corresponds to a setting that is similar to our choice bandits framework, however, the analysis in [17] is in the PAC/pure exploration setting, while ours is in the regret minimization setting.

**Top-$k$ identification under MNL model (PAC setting).** Recently, there has also been work on identifying the top-$k$ items under an MNL model from actively selected sets $S_t$ in the PAC/pure exploration setting [18].

# C    Proof of Lower Bound (Theorem 1)

We say that an algorithm is *strongly consistent* under GCC if its expected regret over $T$ trials is $o(T^a)$ for a constant $a < 1$ under any model in this class.

**Theorem 1.** *Given a set of arms $[n]$, choice set size bound $k \le n$, parameter $\Delta \in (0, 1)$, and any strongly consistent algorithm $\mathcal{A}$ under GCC, there exists a GCC choice model with $\Delta_{\min}^{\text{GCC}} = \Delta$ such that when choice outcomes are drawn from this model we have*

$$\liminf_{T \to \infty} \frac{\mathbf{E}[R(T)]}{\log T} = \Omega\left(\frac{n-1}{\Delta}\right),$$

*where $T$ is the time-horizon. If the underlying model is MNL with parameters $v_1, v_2, \cdots v_n \in \mathbb{R}$, then:*

$$\liminf_{T \to \infty} \frac{\mathbf{E}[R(T)]}{\log T} = \Omega\left(\sum_{i \in [n] \setminus \{i^*\}} \frac{1}{\Delta_{i^* i}^{\text{MNL}}}\right),$$

*where $\Delta_{i^* i}^{\text{MNL}} = \frac{e^{v_{i^*}} - e^{v_i}}{e^{v_{i^*}} + e^{v_i}}$, for $i \in [n] \setminus \{i^*\}$.*

The above theorem states that there exists a model in the GCC class where any strongly consistent algorithm needs to incur $\Omega(n \log T)$ regret. If the underlying model is MNL, then such an algorithm will again incur $\Omega(n \log T)$ regret,

however, we provide a more refined instance-wise bound in this case. Also note the difference in quantifiers 'there exists' for GCC and 'for any' for MNL.

We will prove this theorem using the following change of measure lemma of [20].

**Lemma 3** ([20]). *Consider two multi-armed bandit instances where $A$ is the set of arms, and the two different collections of reward distributions are $\boldsymbol{\mu} = \{\mu_i : \forall i \in A\}$ and $\boldsymbol{\mu}' = \{\mu_i' : \forall i \in A\}$, let $i_t$ be the arm played at trial $t$ by an algorithm and $X_t$ be the reward at time $t$, and let $\mathcal{F}_t = \sigma(i_1, X_1, \cdots, i_t, X_t)$ be the sigma algebra upto time $t$. Consider a $\mathcal{F}_T$ measurable random variable $Z \in [0, 1]$, then*

$$\sum_{i \in A} \mathbf{E}_{\boldsymbol{\mu}}[N_i(T)] KL(\mu_i, \mu_i') \geq d(\mathbf{E}_{\boldsymbol{\mu}}[Z], \mathbf{E}_{\boldsymbol{\mu}'}[Z]),$$

*where $N_i(T)$ denotes the number of pulls of arm $i$ in $T$ trials and $KL$ is the Kullback-Leibler divergence between two distributions, and $d(p; q)$ is the Kullback-Leibler divergence between Bernoulli distributions with parameters $p$ and $q$.*

In the proof of the lower bound we first bound the number of times an arm is played using the above lemma, and then bound the total regret due to this arm. Let us first define the regret per arm $i \in [n]$ as

$$R(T, i) = \sum_{t=1}^{T} \mathbb{1}[i \in S_t] \cdot (P_{i^*|S_t \cup i^*} - P_{i|S_t \cup i^*}).$$

We will now provide the proof of the lower bound.

*Proof of Theorem 1.* Given a $\Delta \in (0, 1)$, we will construct instance $\mathbf{P}$ of the choice bandits problem with $n$ arms such that the GCW arm $i^*$ is arm 1. Under this instance, given any set $S$ such that $i^* \in S$, we have $P_{i^*|S} = \frac{1+\Delta}{|S|(1-\Delta)+2\Delta}$ and for any $i \in S \setminus \{i^*\}$, $P_{i|S} = \frac{1-\Delta}{|S|(1-\Delta)+2\Delta}$. Given any set $S$ such that $i^* \notin S$, we will let an arbitrary chosen arm $i_S^* \in S$ be the arm with the highest choice probability in $S$. We have $P_{i_S^*|S} = \frac{1+\Delta}{|S|(1-\Delta)+2\Delta}$, and for any $i \in S \setminus \{i_S^*\}$, $P_{i|S} = \frac{1-\Delta}{|S|(1-\Delta)+2\Delta}$. Note that $i_S^*$ will be equal to $i^*$ when $i^* \in S$. For any set $S$ with $|S| \geq 2$ and $i \in S$, the instance $\mathbf{P}$ also satisfies that $\frac{3}{2}\left(P_{i^*|S\cup i^*} - P_{i|S\cup i^*}\right) \geq P_{i_S^*|S} - P_{i|S}$.

For $i \in [n] \setminus \{1\}$, we will now modify this instance to create a new instance $\mathbf{P}'$ where the GCW arm is $i$. Now, in the new instance, for any set $S$, we will have that $P'_{i_S^*|S} := P_{i|S}$ and $P'_{i|S} := P_{i_S^*|S}$ and for all $j \in S \setminus \{i_S^*, i\}$ we will have $P'_{j|S} := P_{j|S}$. Clearly, the best arm in this new instance is the arm $i$ as it has the highest choice probability in any choice set. It is also easy to verify that both instances belong to the GCC class.

Now, given any set $S$, the probability distributions $P_S$ and $P'_S$ associated with this set are categorical distributions where the feedback is $j$ with probability $P_{j|S}$ and $P_{j'|S}$, respectively. Now, let $A := \{S \subseteq [n] : |S| \leq k\}$ be the set of choice sets of size at most $k$. We can then use Lemma 3 with arms corresponding to sets in $A$ and the reward for set $S$ being drawn from categorical distributions $P_S$ and $P'_S$. We then have the following bound–

$$\sum_{S \in A} \mathbf{E}_{\mathbf{P}}[N_S(T)] KL(P_S, P'_S) \geq d(\mathbf{E}_{\mathbf{P}}[Z], \mathbf{E}_{\mathbf{P}'}[Z]).$$

where $N_S(T)$ is the number of times set $S$ is played in $T$ rounds, and $Z$ is any $\mathcal{F}_T$ measurable random variable. Also, let $A^i = \{S \in A \setminus \{i\} : i \in S\}$ be all sets that contain $i$ except the singleton set $\{i\}$. Since, we have that for any $S \in A \setminus A^i$ the KL divergence $KL(P_S, P'_S) = 0$, then the above bound becomes:

$$\sum_{S \in A^i} \mathbf{E}_{\mathbf{P}}[N_S(T)] KL(P_S, P'_S) \geq d(\mathbf{E}_{\mathbf{P}}[Z], \mathbf{E}_{\mathbf{P}'}[Z]).$$

Given any set $S \in A^i$ we can now calculate the KL divergence between the two categorical distributions using the

inequality $KL(p, q) \leq \sum_{x \in \mathcal{X}} \frac{(p(x) - q(x))^2}{q(x)}$, where $\mathcal{X}$ is the support of the two distributions.

$$KL(P_S, P'_S) \leq \sum_{j \in S} \frac{(P_{j|S} - P'_{j|S})^2}{P'_{j|S}}$$

$$= \frac{(P_{i|S} - P'_{i|S})^2}{P'_{i|S}} + \frac{(P_{i^*_S|S} - P'_{i^*_S|S})^2}{P'_{i^*_S|S}}$$

$$= \frac{(P_{i|S} - P_{i^*_S|S})^2}{P_{i^*_S|S}} + \frac{(P_{i|S} - P_{i^*_S|S})^2}{P_{i|S}}$$

Now, similar to [8], let $Z$ be the fraction of times out of $T$ the singleton set $\{i\}$ is played, i.e. $Z = N_i(T)/T$ where $N_i(T)$ counts the number of times set $\{i\}$ is played. We will then have

$$d(\mathbf{E_P}[Z], \mathbf{E_{P'}}[Z]) \geq \left(1 - \frac{\mathbf{E_P}[N_i(T)]}{T}\right) \ln \frac{T}{T - \mathbf{E_{P'}}[N_i(T)]} - \ln 2 \,.$$

Since, the algorithm is strongly consistent it can only play a suboptimal arm $\{i\}$ only a sublinear number of times, i.e. $\mathbf{E_P}[N_i(T)] = o(T^\alpha)$ and $T - \mathbf{E_{P'}}[N_i(T)] = o(T^\alpha)$ for some $\alpha < 1$. Hence, we have that

$$\lim_{T \to \infty} \frac{1}{\ln T} d(\mathbf{E_P}[Z], \mathbf{E_{P'}}[Z]) \geq \lim_{T \to \infty} \frac{1}{\ln T}\left(1 - \frac{o(T^\alpha)}{T}\right) \ln \frac{T}{o(T^\alpha)} - \ln 2 \geq (1 - \alpha). \tag{C.1}$$

Combining this with the previous inequality, we have that

$$\lim_{T \to \infty} \frac{1}{\ln T} \sum_{S \in A^i} \mathbf{E_P}[N_S(T)]\left(\frac{(P_{i|S} - P_{i^*_S|S})^2}{P_{i^*_S|S}} + \frac{(P_{i|S} - P_{i^*_S|S})^2}{P_{i|S}}\right) \geq (1 - \alpha)$$

$$\implies \lim_{T \to \infty} \frac{1}{\ln T} \sum_{S \in A^i} \mathbf{E_P}[N_S(T)] \cdot (P_{i|S} - P_{i^*_S|S})\left(\frac{(P_{i|S} - P_{i^*_S|S})}{P_{i^*_S|S}} + \frac{(P_{i|S} - P_{i^*_S|S})}{P_{i|S}}\right) \geq (1 - \alpha)$$

$$\implies \lim_{T \to \infty} \frac{1}{\ln T} \sum_{S \in A^i} \mathbf{E_P}[N_S(T)] \cdot \frac{3}{2} \cdot (P_{i^*|S \cup i^*} - P_{i|S \cup i^*})\left(\frac{(P_{i^*_S|S} - P_{i|S})}{P_{i^*_S|S}} + \frac{(P_{i^*_S|S} - P_{i|S})}{P_{i|S}}\right) \geq (1 - \alpha)$$

$$\lim_{T \to \infty} \frac{1}{\ln T} \mathbf{E}[R(T, i)] \cdot \frac{3}{2}\left(\frac{(P_{i^*_S|S} - P_{i|S})}{P_{i^*_S|S}} + \frac{(P_{i^*_S|S} - P_{i|S})}{P_{i|S}}\right) \geq (1 - \alpha),$$

where the second last equation follows from the properties of the underlying instance, and the last equation follows from the definition of regret per arm. We will now argue that

$$\left(\frac{(P_{i^*_S|S} - P_{i|S})}{P_{i^*_S|S}} + \frac{(P_{i^*_S|S} - P_{i|S})}{P_{i|S}}\right) = \frac{2\Delta}{1 + \Delta} + \frac{2\Delta}{1 - \Delta} = \frac{4\Delta}{(1 + \Delta)(1 - \Delta)} \,.$$

Using this we will have that

$$\lim_{T \to \infty} \frac{1}{\ln T} \mathbf{E}[R(T, i)] \cdot \frac{3}{2}\left(\frac{(P_{i^*_S|S} - P_{i|S})}{P_{i^*_S|S}} + \frac{(P_{i^*_S|S} - P_{i|S})}{P_{i|S}}\right) \geq (1 - \alpha)$$

$$\implies \lim_{T \to \infty} \frac{1}{\ln T} \mathbf{E}[R(T, i)] \cdot \frac{4\Delta}{(1 + \Delta)(1 - \Delta)} \geq (1 - \alpha) \cdot \frac{2}{3}$$

$$\implies \lim_{T \to \infty} \frac{1}{\ln T} \mathbf{E}[R(T, i)] \geq (1 - \alpha) \cdot \frac{(1 + \Delta)(1 - \Delta)}{4\Delta} \cdot \frac{2}{3} \,.$$

We also have that $\frac{(1+\Delta)(1-\Delta)}{4\Delta} = \Omega(\frac{1}{\Delta})$ for any $\Delta$ bounded away from 1. Since, we have that $R(T) = \sum_{i \in [n]} R(T, i)$ we get that

$$\lim_{T \to \infty} \frac{1}{\ln T} \mathbf{E}[R(T)] = \Omega\left(\frac{n-1}{\Delta}\right),$$

which concludes the proof of the lower bound for the general GCC class.

Now, given any MNL instance, we also derive a regret lower bound which gives the minimum instance-wise regret any strongly-consistent algorithm for the GCC class needs to incur under this MNL instance.

Consider an instance $\mathbf{P}$ with an underlying MNL model with weights $v_1, \cdots, v_n$. We will assume that all these weights are distinct for simplicity, otherwise we can add a small perturbation to these weights to break ties. We will re-parameterize this instance, and let $w_i := \log v_i$ for any $i \in [n]$. Given any set $S$, let $w_S = \sum_{j \in [n]} w_j$. We have that $P_{i|S} = w_i/w_S$ for any $i \in S$. Given $S$, we will again let $i_S^*$ to be the arm that has the highest choice probability in $S$, i.e. $i_S^* = \operatorname{argmax}_{i \in S} w_i$. We will denote by $\kappa$ the ratio of the maximum weight to minimum weight, i.e. $\kappa = \max_i w_i / \min_j w_j$.

For $i \in [n] \setminus \{1\}$, we will now modify this instance to create a new instance $\mathbf{P}'$ where the GCW arm is $i$. In the new instance, for any set $S$, we will have that $P'_{i_S^*|S} := P_{i|S}$ and $P'_{i|S} := P_{i_S^*|S}$ and for all $j \in S \setminus \{i_S^*, i\}$ we will have $P'_{j|S} := P_{j|S}$. Clearly, the best arm in this new instance is the arm $i$ as it has the highest choice probability in any choice set. It is also easy to verify that this new instance $\mathbf{P}'$ belongs to the GCC class. Note that $\mathbf{P}'$ might not belong to the MNL class. Under the instance $\mathbf{P}$ we have that $(1 + \kappa)(P_{i^*|S \cup i^*} - P_{i|S \cup i^*}) \geq (P_{i_S^*|S} - P_{i|S})$.

Given these two instances, we can follow steps analogous to the proof of the GCC case, to derive the following bound

$$\lim_{T \to \infty} \frac{1}{\ln T} \mathbf{E}[R(T, i)] \cdot (1 + \kappa) \left(\frac{(P_{i_S^*|S} - P_{i|S})}{P_{i_S^*|S}} + \frac{(P_{i_S^*|S} - P_{i|S})}{P_{i|S}}\right) \geq (1 - \alpha).$$

We now have that

$$\left(\frac{(P_{i_S^*|S} - P_{i|S})}{P_{i_S^*|S}} + \frac{(P_{i_S^*|S} - P_{i|S})}{P_{i|S}}\right) = \frac{w_{i_S^*} - w_i}{w_i} + \frac{w_{i_S^*} - w_i}{w_{i_S^*}} = \frac{w_{i_S^*} - w_i}{w_{i_S^*} + w_i}\left(\frac{w_{i_S^*} + w_i}{w_i} + \frac{w_{i_S^*} + w_i}{w_{i_S^*}}\right)$$

$$\leq \frac{w_{i^*} - w_i}{w_{i^*} + w_i}(3 + \kappa) = \Delta_{i^*i}^{\mathrm{MNL}}(3 + \kappa)$$

Using the same steps as above we have that

$$\lim_{T \to \infty} \frac{1}{\ln T} \mathbf{E}[R(T, i)] \geq (1 - \alpha) \cdot \frac{1}{\Delta_{i^*i}^{\mathrm{MNL}}} \cdot \frac{1}{(3 + \kappa)(1 + \kappa)}.$$

Since, we have that $R(T) = \sum_{i \in [n]} R(T, i)$ we get that

$$\lim_{T \to \infty} \frac{1}{\ln T} \mathbf{E}[R(T)] = \Omega\left(\sum_{i \in [n] \setminus \{i^*\}} \frac{1}{\Delta_{i^*i}^{\mathrm{MNL}}}\right),$$

which concludes the proof of the lower bound for the MNL case.

$\square$

Note that the lower bound for the MNL model also implies a lower bound for the general GCC class. However, we chose to construct an instance outside MNL for the GCC lower bound in order to show that such a lower bound also holds beyond the MNL. Also, note that the lower bound in [8] for MNL under MNL consistent algorithms is worst-case while our lower bound for MNL under GCC consistent algorithms applies to all MNL instances.

# D A Concentration Inequality for Pairwise Estimates

In this section we will prove our concentration inequality that would be needed to bound the deviation in the pairwise preference estimates extracted from multiway comparisons.

**Lemma 2.** *Consider a GCC choice model with GCW $i^*$. Fix $i \in [n]$. Let $S_1, \cdots, S_T$ be a sequence of subsets of $[n]$ and $y_1, \cdots, y_T$ be a sequence of choices according to this model, let $\mathcal{F}_t = \{S_1, y_1, \cdots, S_t, y_t\}$ be a filtration containing the history of execution of the algorithm such that $S_{t+1}$ is a measurable function of $\mathcal{F}_t$. Let $\hat{P}_{i^*i}(t)$ be the empirical probability estimate of $i^*$ beating $i$ calculated according to Equation 4.1, then for any given $t \in [T]$ we have that*

$$\Pr\Big(\hat{P}_{i^*i}(t) \leq P_{i^*i}^{\mathrm{GCC}} - \epsilon \text{ and } N_{i^*i}(t) \geq m\Big) \leq e^{-d(P_{i^*i}^{\mathrm{GCC}} - \epsilon, P_{i^*i}^{\mathrm{GCC}}) \cdot m} \tag{D.1}$$

*where*

$$P_{i^*i}^{\mathrm{GCC}} = \min_{S : |S| \leq k, \{i^*, i\} \subseteq S} \frac{P_{i^*|S}}{P_{i^*|S} + P_{i|S}}, \tag{D.2}$$

*and $d(\cdot, \cdot)$ is the KL-divergence between two Bernoulli distributions, and $N_{i^*i}(t) := \sum_{t'=1}^{t} \mathbb{1}(a_{t'} = i, \{i^*, i\} \subseteq S_{t'}, y_{t'} \in \{i^*, i\})$. The above bound implies the following bound*

$$\Pr\bigg(\hat{P}_{i^*i}(t) \leq \frac{1}{2}; N_{i^*i}(t) \geq m\bigg) \leq e^{-d(\frac{1}{2}, P_{i^*i}^{\mathrm{GCC}})m} \tag{D.3}$$

*We also have the following bound–*

$$\Pr\Big(\hat{P}_{ii^*}(t) \geq P_{ii^*}^{\mathrm{GCC}} + \epsilon; N_{i^*i}(t) \geq m\Big) \leq e^{-d(P_{i^*i}^{\mathrm{GCC}} - \epsilon, P_{i^*i}^{\mathrm{GCC}}) \cdot m} \tag{D.4}$$

*where $P_{ii^*}^{\mathrm{GCC}} = 1 - P_{i^*i}^{\mathrm{GCC}}$.*

*Proof.* We will first prove inequality D.1. Let $Z_1, Z_2, \cdots$ be a sequence of i.i.d. Bernoulli random variables with probability of success $P_{i^*i}^{\mathrm{GCC}}$. We will initialize a counter $C$ to 0. Let us consider an alternate process for generating multiway choices $y'_t$ from sets $S_t$. In this process, given any $t$ and a set $S_t$ such that $i^*, i \in S_t$ with $a_t = i$, we first generate a Bernoulli random variable $X_t$ with probability $P_{i^*|S} + P_{i|S}$. If $X_t = 0$ we sample a multinomial random variable $Y_t$ such that $Y_t = j$ with probability $\frac{P_{j|S}}{1 - P_{i^*|S} - P_{i|S}}$, for $j \in S \setminus \{i, i^*\}$, and let $y'_t = Y_t$. If $X_t = 1$, then we increase the counter $C$ by 1, and sample the Bernoulli random variable $Z_C$ with probability $P_{i^*i}^{\mathrm{GCC}}$. If $Z_C = 1$ we declare $i^*$ as the choice, i.e. $y'_t = i^*$, otherwise if $Z_C = 0$ we declare $i$ to be the choice. Let $P_{i^*i|S} = P_{i^*|S}/(P_{i^*|S} + P_{i|S})$. Now, we couple the process generating $y'_t$ and the process generating $y_t$ as follows: if $y'_t \in S_t \setminus \{i\}$ then we let $y_t = y'_t$, otherwise if $y'_t = i$ then we let $y_t = i^*$ with probability $(P_{i^*i|S_t} - P_{i^*i}^{\mathrm{GCC}})/(1 - P_{i^*i}^{\mathrm{GCC}})$ and let $y_t = i$ with probability $(1 - P_{i^*i|S_t})/(1 - P_{i^*i}^{\mathrm{GCC}})$. The first thing to check is that $y_t$ is drawn from the correct probabilities $P_{y_t|S_t}$ according to the underlying choice model. We have, for any $j \in S_t \setminus \{i^*, i\}$

$$\begin{aligned}
\Pr\{y_t = j | S_t\} &= \Pr\{X_t = 0, Y_t = j | S_t\} \\
&= \Pr\{X_t = 0 | S_t\} \Pr\{Y_t = j | X_t = 0, S_t\} \\
&= \big(1 - P_{i^*|S_t} - P_{i|S_t}\big) \cdot \frac{P_{j|S_t}}{1 - P_{i^*|S_t} - P_{i|S_t}} \\
&= P_{j|S_t}
\end{aligned}$$

We also have that

$$\Pr\{y_t = i^*|S_t\} = \Pr\{X_t = 1, Y_t = i^*|S_t\} + \frac{P_{i^*i|S_t} - P_{i^*i}^{\text{GCC}}}{1 - P_{i^*i}^{\text{GCC}}} \cdot \Pr\{X_t = 1, Y_t = i|S_t\}$$

$$= \left(P_{i^*|S_t} + P_{i|S_t}\right) \cdot \left(P_{i^*i}^{\text{GCC}} + (1 - P_{*i}^{\text{GCC}}) \cdot \frac{P_{i^*i|S_t} - P_{i^*i}^{\text{GCC}}}{1 - P_{i^*i}^{\text{GCC}}}\right)$$

$$= \left(P_{i^*|S_t} + P_{i|S_t}\right) \cdot \left(P_{i^*i|S_t}\right)$$

$$= P_{i^*|S_t}$$

where the last inequality follows from definition of $P_{i^*i|S}$. The fact that $\Pr\{y_t = i|S_t\} = P_{i|S}$ follows from the fact that the choice probabilities sum to 1.

Let $W_{i^*i}(t) = \sum_{t'=1}^{t} \mathbb{1}(a_{t'} = i, \{i^*, i\} \subseteq S_{t'}, y_{t'} = i^*)$ and $W'_{i^*i}(t) = \sum_{t'=1}^{t} \mathbb{1}(a_{t'} = i, \{i^*, i\} \subseteq S_{t'}, y'_{t'} = i^*)$. Due to the above coupling, we immediately have that $\Pr(W_{i^*i}(t)) \geq \Pr(W'_{i^*i}(t))$ for any $t \in [T]$. Then

$$\Pr(W_{i^*i}(t) \leq r) \leq \Pr(W'_{i^*i}(t) \leq r)$$

for any $r \geq 0$, and any $t \in [T]$. Using this, we have that

$$\Pr\left(\hat{P}_{i^*i}(t) \leq P_{i^*i}^{\text{GCC}} - \epsilon; N_{i^*i}(t) \geq m\right) = \Pr\left(W_{i^*i}(t) \leq N_{i^*i}(t) \cdot (P_{i^*i}^{\text{GCC}} - \epsilon); N_{i^*i}(t) \geq m\right)$$

$$\leq \Pr\left(W'_{i^*i}(t) \leq N_{i^*i}(t) \cdot (P_{i^*i}^{\text{GCC}} - \epsilon); N_{i^*i}(t) \geq m\right)$$

Now, using techniques similar to [21], we have the following bound

$$\Pr\left(\frac{W'_{i^*i}(t)}{N_{i^*i}(t)} \leq P_{i^*i}^{\text{GCC}} - \epsilon; N_{i^*i}(t) \geq m\right) = \Pr\left(\frac{\sum_{s=1}^{N_{i^*i}(t)} Z_s}{N_{i^*i}(t)} \leq P_{i^*i}^{\text{GCC}} - \epsilon; N_{i^*i}(t) \geq m\right)$$

$$= \sum_{r=m}^{t} \Pr\left(\frac{\sum_{s=1}^{r} Z_s}{r} \leq P_{i^*i}^{\text{GCC}} - \epsilon; N_{i^*i}(t) = r\right)$$

$$= \sum_{r=m}^{t} \Pr\left(\frac{\sum_{s=1}^{r} Z_s}{r} \leq P_{i^*i}^{\text{GCC}} - \epsilon\right) \Pr(N_{i^*i}(t) = r)$$

where the last equality holds because of the fact that $Z_1, Z_2, \cdots$ is an independent sequence of random variables that do not lie in the sigma algebra of $S_1, \cdots, S_t, X_1, \cdots, X_t$. Using the KL-divergence based concentration inequality from [22] we have that

$$\Pr\left(\frac{\sum_{s=1}^{r} Z_s}{r} \leq P_{i^*i}^{\text{GCC}} - \epsilon\right) \leq e^{-d(P_{i^*i}^{\text{GCC}} - \epsilon, P_{i^*i}^{\text{GCC}})r}.$$

We then have that

$$\sum_{r=m}^{t} \Pr\left(\frac{\sum_{s=1}^{r} Z_s}{r} \leq P_{i^*i}^{\text{GCC}} - \epsilon\right) \Pr(N_{i^*i}(t) = r) \leq \sum_{r=m}^{t} e^{d(P_{i^*i}^{\text{GCC}} - \epsilon, P_{i^*i}^{\text{GCC}})r} \Pr(N_{i^*i}(t) = r) \leq e^{-d(P_{i^*i}^{\text{GCC}} - \epsilon, P_{i^*i}^{\text{GCC}})m}$$

The proof of reverse direction follows from a similar coupling argument followed by the above concentration inequality. $\square$

Note that the above coupling technique has similarity to the coupling used in [21] in order to show concentration of pairwise estimates under the MNL model. However, this argument relies on the IIA property of MNL, which does not hold under general GCC models.

# E   Proof of Regret Bound for WBA

In this section we will prove the regret bound for our WBA algorithm. The following theorem presents the bound.

**Theorem 2.** *Let $n$ be the number of arms, $k \leq n$ be the choice set size parameter, and $i^*$ be the GCW arm . If the multiway choices are drawn according to a GCC choice model with $\Delta_{\min}^{\text{GCC}}$ and $\Delta_{\max}^{\text{GCC}}$ defined in Equation 3.1, then for any $C \geq 1/(\Delta_{\min}^{\text{GCC}})^4$, the expected regret incurred by WBA is upper bounded by*

$$\mathbf{E}\left[R(T)\right] \leq O\left(\frac{n^2 \log n}{(\Delta_{\min}^{\text{GCC}})^2}\right) + O\left(n \log(TC) \cdot \frac{\Delta_{\max}^{\text{GCC}}}{(\Delta_{\min}^{\text{GCC}})^2}\right),$$

*where $T$ is the (unknown) time-horizon. If the underlying model is MNL with weights $v_1, \cdots, v_n \in \mathbb{R}$, then for any $C \geq 1/(\Delta_{\min}^{\text{MNL}})^4$, we have*

$$\mathbf{E}\left[R(T)\right] \leq O\left(\frac{n^2 \log n}{(\Delta_{\min}^{\text{MNL}})^2}\right) + O\left(\sum_{i \in [n] \setminus i^*} \frac{\log(TC)}{\Delta_{i^*i}^{\text{MNL}}}\right),$$

*where $\Delta_{i^*i}^{\text{MNL}} = \frac{e^{v_{i^*}} - e^{v_i}}{e^{v_{i^*}} + e^{v_i}}$ and $\Delta_{\min}^{\text{MNL}} := \min_{i \neq i^*} \Delta_{i^*i}^{\text{MNL}}$.*

The proof of the above theorem hinges on three main lemmas given below. Before stating these lemmas, we would like to remind the reader that the execution of our algorithm is divided in rounds and each round contain up to $n$ trials. The first lemma bounds the number of rounds arm $i^*$ is not in the active set.

**Lemma 4** (Number of rounds where $i^*$ is not active). *Fix an anchor arm $a \in [n] \setminus \{i^*\}$. The expected number of rounds arm $i^*$ will not be a part of the active set is bounded as*

$$\mathbf{E}\left[\sum_{r=1}^{T} \mathbb{1}[i^* \notin A_r]\right] \leq 2.$$

We will define $a_r$ to be the arm that empirically beats all other arms at the end of round $r-1$ if such an arm exists, i.e. $\sum_{j \in [n]} \mathbb{1}[\hat{P}_{j a_r}(t) \leq \frac{1}{2}] = n - 1$, where $t$ is the last trial in round $r-1$. If there is no arm that empirically beats all other arms then we will let $a_r = 0$. If there are multiple such arms, then we will choose one arbitrarily. The following lemma will now bound the number of rounds arm $i^*$ does not empirically beat every other arm.

**Lemma 5** (Time when $i^*$ is not the empirically best arm). *The total number of rounds when the best arm $i^*$ will not be the empirically best arm, even when it is in the active set, is upper bounded as*

$$\mathbf{E}\left[\sum_{r=1}^{T} \mathbb{1}[a_r \neq i^*, i^* \in A_r]\right] \leq \sum_{i \in [n] \setminus \{i^*\}} \frac{1}{\exp\{d(1/2, P_{i^*i}^{\text{GCC}})\} - 1},$$

*where $P_{i^*i}^{\text{GCC}}$ is defined in Equation D.2.*

Note that if $a_r = i^*$ then the anchor arm in all the trials in that round becomes $i^*$. Let us define the regret per arm $i \in [n]$ for a set $S$ as

$$r(S, i) = \mathbb{1}[i \in S] \cdot (P_{i^*|S \cup i^*} - P_{i|S \cup i^*}).$$

The following lemma now bounds the regret incurred due to each suboptimal arm when played against the anchor $i^*$.

**Lemma 6** (Regret due to a bad arm). *Given an arm $i \in [n] \setminus \{i^*\}$ the expected regret incurred due to arm $i$ when arm $i^*$ is the anchor is upper bounded as*

$$\mathbf{E}\left[\sum_{t=1}^{T} r(S_t, i) \cdot \mathbb{1}[a_t = i^*, i \in S_t]\right] \leq \Delta_{i^*i}^{\text{GCC}} \cdot \frac{2e}{e-1} \cdot \left(\frac{(1+\delta) \log(TnC)}{d(P_{i^*i}^{\text{GCC}}, \frac{1}{2})} + \frac{1}{\Omega(\delta^2)}\right),$$

*where $\delta > 0$ is some constant, and $\Delta_{i^*i}^{\text{GCC}} = \max_{S:|S| \leq k} \Delta_{i^*i|S}$.*

We will now prove the above theorem using the three lemmas above.

*Proof of Theorem 2.* The execution of the algorithm can roughly be divided into three intermittent phases– (1) when the GCW arm $i^*$ is not in the active set, (2) when $i^*$ is in the active set but does not beat all other arms empirically, i.e. $a_r \neq i^*$, (3) when $i^*$ is in the active set and also beats all other arms empirically. The three lemmas above bound the number of rounds spent in these three phases.

However, in order to prove a regret upper bound we will also have to bound the total regret incurred due to a single round. The first thing to observe is that each arm is played at most once in each round except a few arms that might be played multiple times due to step 6 of the algorithm. Hence, the regret for all steps except step 6 is upper bounded by $n$ as the regret for each arm is at most 1. Now, in order to bound the regret for step 6, we need to observe that the number of times the anchor arm is changed in a single round can be at most $\log n$. This is due to the fact that $A_r \setminus Q$ reduces by a factor of at least 2 each time a new anchor arm is selected by the algorithm. Now, we can bound the regret incurred due to step 6 of the algorithm by $k \log n \leq n \log n$ as the regret for each arm is upper bounded by 1 and there can be at most $k$ arms added in step 6 per anchor arm.

Hence, we now have that

$$\mathbf{E}[R(T)] \leq n \log n \cdot \left( \mathbf{E}\left[ \sum_{r=1}^{T} \mathbb{1}[i^* \notin A_r] \right] + \mathbf{E}\left[ \sum_{r=1}^{T} \mathbb{1}[a_r \neq i^*, i^* \in A_r] \right] \right) + \sum_{i \in [n] \setminus \{i^*\}} \mathbf{E}\left[ \sum_{t=1}^{T} r(S_t, i) \cdot \mathbb{1}[a_t = i^*, i \in S_t] \right]$$

$$\leq n \log n \cdot \left( 2 + \sum_{i \in [n] \setminus \{i^*\}} \frac{1}{\exp\{d(1/2, P_{i^*i}^{\mathrm{GCC}})\} - 1} \right) + \sum_{i \in [n] \setminus \{i^*\}} \Delta_{i^*i}^{\mathrm{GCC}} \cdot \frac{2e}{e-1} \cdot \left( \frac{(1+\delta) \log(TnC)}{d(P_{i^*i}^{\mathrm{GCC}}, \frac{1}{2})} + \frac{1}{\Omega(\delta^2)} \right)$$

$$\leq O\left( \frac{n^2 \log n}{(\Delta_{\min}^{\mathrm{GCC}})^2} \right) + n \cdot \Delta_{\max}^{\mathrm{GCC}} \cdot \frac{2e}{e-1} \cdot \frac{1}{\Omega(\delta^2)} + \sum_{i \in [n] \setminus \{i^*\}} \Delta_{i^*i}^{\mathrm{GCC}} \cdot \frac{2e}{e-1} \cdot \frac{(1+\delta) \log(TnC)}{d(P_{i^*i}^{\mathrm{GCC}}, \frac{1}{2})}$$

$$= O\left( \frac{n^2 \log n}{(\Delta_{\min}^{\mathrm{GCC}})^2} \right) + \sum_{i \in [n] \setminus \{i^*\}} \Delta_{i^*i}^{\mathrm{GCC}} \cdot \frac{2e}{e-1} \cdot \frac{(1+\delta) \log(TC)}{d(P_{i^*i}^{\mathrm{GCC}}, \frac{1}{2})}$$

where the third inequality follows from the well-known Pinsker's inequality $d(P, Q) \geq 2(P - Q)^2$ and the last inequality holds for any constant $\delta$. Now again using the Pinsker's inequality we have that $d(P_{i^*i}^{\mathrm{GCC}}, \frac{1}{2}) \geq (\Delta_{\min}^{\mathrm{GCC}})^2/2$. For a general GCC model, we then have that

$$\mathbf{E}[R(T)] \leq O\left( \frac{n^2 \log n}{(\Delta_{\min}^{\mathrm{GCC}})^2} \right) + \sum_{i \in [n] \setminus \{i^*\}} \Delta_{i^*i}^{\mathrm{GCC}} \cdot \frac{4e}{e-1} \cdot \frac{(1+\delta) \log(TC)}{\Delta_{\min}^{\mathrm{GCC}}}$$

$$\leq O\left( \frac{n^2 \log n}{(\Delta_{\min}^{\mathrm{GCC}})^2} \right) + n \cdot \Delta_{\max}^{\mathrm{GCC}} \cdot \frac{4e}{e-1} \cdot \frac{(1+\delta) \log(TC)}{(\Delta_{\min}^{\mathrm{GCC}})^2}$$

which gives the desired bound under any GCC model.

Now, if the underlying GCC model is MNL, then we have $d(P_{i^*i}^{\mathrm{GCC}}, \frac{1}{2}) \geq (\Delta_{i^*i}^{\mathrm{MNL}})^2/2$ and $\Delta_{i^*i}^{\mathrm{GCC}} = \Delta_{i^*i}^{\mathrm{MNL}}$. We then have that

$$\mathbf{E}[R(T)] \leq O\left( \frac{n^2 \log n}{(\Delta_{\min}^{\mathrm{MNL}})^2} \right) + \sum_{i \in [n] \setminus \{i^*\}} \Delta_{i^*i}^{\mathrm{MNL}} \cdot \frac{4e}{e-1} \cdot \frac{(1+\delta) \log(TC)}{(\Delta_{i^*i}^{\mathrm{MNL}})^2}$$

$$= O\left( \frac{n^2 \log n}{(\Delta_{\min}^{\mathrm{MNL}})^2} \right) + \sum_{i \in [n] \setminus \{i^*\}} \frac{4e}{e-1} \cdot \frac{(1+\delta) \log(TC)}{\Delta_{i^*i}^{\mathrm{MNL}}} .$$

$\square$

## E.1 Proof of Lemma 4

The following lemma calculates the expected number of rounds arm $i^*$ will not be played.

**Lemma 4** (Number of rounds where $i^*$ is not active). *Fix an anchor arm $a \in [n] \setminus \{i^*\}$. The expected number of rounds arm $i^*$ will not be a part of the active set is bounded as*

$$\mathbf{E}\left[\sum_{r=1}^{T} \mathbb{1}[i^* \notin A_r]\right] \leq 2 .$$

*Proof.* We have that

$$\mathbf{E}\left[\sum_{r=1}^{T} \mathbb{1}[i^* \notin A_r]\right] = \mathbf{E}\left[\sum_{r=2}^{T} \mathbb{1}[i^* \notin A_r]\right] \leq \mathbf{E}\left[\sum_{t=2}^{T} \mathbb{1}[\neg \mathcal{J}_{i^*}(t, C)]\right] .$$

The first equality above follows due to the fact that $A_1$ will always include $i^*$. Using the union bound we have the following inequality-

$$\mathbb{1}[\neg \mathcal{J}_{i^*}(t, C)] \leq \sum_{S \subseteq [n] \setminus \{i^*\}} \sum \sum \cdots \sum_{\{n_a\} \in [T]^S}$$

$$\mathbb{1}[\bigcap_{a \in S}\{N_{i^*a}(t) = n_a, \hat{P}_{i^*a}(t) < \frac{1}{2}\} \cap \bigcap_{a \notin S}\{\hat{P}_{i^*a}(t) \geq \frac{1}{2}\} \cap \{\neg \mathcal{J}_{i^*}(t, C)\}] .$$

Fix some set $S \subseteq [n] \setminus \{i^*\}$. Also, let $s := |S|$. Fix some $n_a \in [T]$ for all $a \in S$. Let $\hat{P}_{i^*a}^{n_a}$ be the empirical probability of $i^*$ beating $a$ after being pulled together $n_a$ times. We will analyze the number of rounds that $i^*$ is excluded from the active set due to the above configuration of $S, \{n_a\}$. The conditions $\mathcal{J}_{i^*}(t, C)$ will hold when

$$\sum_{a \in S} n_a d(\hat{P}_{i^*a}^{n_a}, \frac{1}{2}) \leq \log(t) + s \log(nC) \implies t \geq \exp\left(\sum_{a \in S} n_a d(\hat{P}_{i^*a}^{n_a}, \frac{1}{2}) - s \log(nC)\right) .$$

Hence, we have that

$$\sum_{t=2}^{\infty} \mathbb{1}[\bigcap_{a \in S}\{N_{i^*a}(t) = n_a, \hat{P}_{i^*a}(t) < \frac{1}{2}\} \cap \bigcap_{a \notin S}\{\hat{P}_{i^*a}(t) \geq \frac{1}{2}\} \cap \{\neg \mathcal{J}_{i^*}(t, C)\}]$$

$$\leq \exp\left(\sum_{a \in S} n_a d(\hat{P}_{i^*a}^{n_a}, \frac{1}{2}) - s \log(nC)\right) .$$

Now, we will use the method similar to the one used in Lemma 5 of [23], to bound the expectation of the above quantity. Fix $x_a \in [0, \log 2]$ for all $a \in S$. Let $P_a(x_a) = \Pr\left(\hat{P}_{i^*a}^{n_a} \leq \frac{1}{2}, d^+(\hat{P}_{i^*a}^{n_a}, \frac{1}{2}) \geq x_a\right)$, where $d^+(P, Q) =$

$\mathbb{1}[P \leq Q] \cdot d(P, Q)$. We then have

$$\mathbf{E}\left[\sum_{t=2}^{T} \mathbb{1}[\bigcap_{a \in S}\{N_{i^*a}(t) = n_a, \hat{P}_{i^*a}(t) < \frac{1}{2}\} \cap \bigcap_{a \notin S}\{\hat{P}_{i^*a}(t) \geq \frac{1}{2}\} \cap \{\neg \mathcal{J}_{i^*}(t, C)\}]\right]$$

$$\leq \int_{\{x_a\} \in [0, \log(2)]^{|S|}} \exp\left(\sum_{a \in S} n_a x_a - s \log(nC)\right) \prod_{a \in S} \mathrm{d}(-P_a(x_a))$$

$$= \exp\left(-s \log(nC)\right) \cdot \prod_{a \in S} \int_{x_a \in [0, \log(2)]} \exp\left(n_a x_a\right) \mathrm{d}(-P_a(x_a))$$

<div align="center">(due to the independence of comparisons with respect to different anchors)</div>

$$= \exp\left(-s \log(nC)\right) \cdot \prod_{a \in S}\left([-\exp(n_a x_a) P_a(x_a)]_0^{\log(2)} + \int_{x_a \in [0, \log(2)]} n_a \exp\left(n_a x_a\right) P_a(x_a) \mathrm{d}x_a\right)$$

<div align="center">(integration by parts)</div>

$$\leq \exp\left(-s \log(nC)\right) \cdot \prod_{a \in S}\left(P_a(0) + \int_{x_a \in [0, \log(2)]} n_a \exp\left(n_a x_a\right) \exp\left\{-n_a(x_a + C_1(P_{i^*a}^{\mathrm{GCC}}, \frac{1}{2}))\right\} \mathrm{d}x_a\right)$$

<div align="center">(Using concentration inequality (Lemma 2) and Fact 10 in [23], with $C_1(p, q) = (p - q)^2/2p(1 - q)$)</div>

$$= \exp\left(-s \log(nC)\right) \cdot \prod_{a \in S}\left(\exp\left\{-n_a d(\frac{1}{2}, P_{i^*a}^{\mathrm{GCC}})\right\} + \int_{x_a \in [0, \log(2)]} n_a \exp\left\{-n_a C_1(P_{i^*a}^{\mathrm{GCC}}, \frac{1}{2})\right\} \mathrm{d}x_a\right)$$

$$= \exp\left(-s \log(nC)\right) \cdot \prod_{a \in S}\left(\exp\left\{-n_a d(\frac{1}{2}, P_{i^*a}^{\mathrm{GCC}})\right\} + \log(2) n_a \exp\left\{-n_a C_1(P_{i^*a}^{\mathrm{GCC}}, \frac{1}{2})\right\}\right).$$

We will now take a union bound over $\{n_a\}$. We have that

$$\sum_{\{n_a\} \in [T]^S}\sum\cdots\sum \exp\left(-s \log(nC)\right) \cdot \prod_{a \in S}\left(\exp\left\{-n_a d(\frac{1}{2}, P_{i^*a}^{\mathrm{GCC}})\right\} + \log(2) n_a \exp\left\{-n_a C_1(P_{i^*a}^{\mathrm{GCC}}, \frac{1}{2})\right\}\right)$$

$$= \exp\left(-s \log(nC)\right) \cdot \prod_{a \in S}\sum_{n_a}\left(\exp\left\{-n_a d(\frac{1}{2}, P_{i^*a}^{\mathrm{GCC}})\right\} + \log(2) n_a \exp\left\{-n_a C_1(P_{i^*a}^{\mathrm{GCC}}, \frac{1}{2})\right\}\right)$$

$$\leq \exp\{-s \log(nC)\} \cdot \prod_{a \in S}\left(\frac{1}{\exp\{d(\frac{1}{2}, P_{i^*a}^{\mathrm{GCC}})\} - 1} + \frac{\exp\{C_1(P_{i^*a}^{\mathrm{GCC}}, \frac{1}{2})\}}{(\exp\{C_1(P_{i^*a}^{\mathrm{GCC}}, \frac{1}{2})\} - 1)^2}\right)$$

$$\leq \exp\{-s \log(nC) + s \log(C')\},$$

where the constant $C'$ is defined as

$$C' := \max_{a \in [n] \setminus i^*}\left(\frac{1}{\exp\{d(\frac{1}{2}, P_{i^*a}^{\mathrm{GCC}})\} - 1} + \frac{\exp\{C_1(P_{i^*a}^{\mathrm{GCC}}, \frac{1}{2})\}}{(\exp\{C_1(P_{i^*a}^{\mathrm{GCC}}, \frac{1}{2})\} - 1)^2}\right) \leq \frac{1}{(\Delta_{\min}^{\mathrm{GCC}})^4}.$$

We will now apply the union bound over all subsets $S \subseteq [n] \setminus i^*$. Now, if the parameter $C$ is larger than $C'$, then we

have

$$\sum_{S \subseteq [n] \setminus \{i^*\}} \exp\{-|S| \log(nC) + |S| \log(C')\} = \sum_{s=1}^{n-1} \sum_{S \subseteq [n] \setminus \{i^*\}, |S|=s} \exp\{-s \log(nC) + s \log(C')\}$$

$$\leq \sum_{s=1}^{n-1} \left(\frac{en}{s}\right)^s \exp\{-s \log(nC) + s \log(C')\}$$

$$= \sum_{s=1}^{n-1} \exp\{-s \log(nC) + s \log(C') + s \log(n) + s - s \log(s)\}$$

$$\leq \sum_{s=1}^{n-1} \exp\{s - s \log(s)\} \leq 2 \,.$$

□

## E.2 Proof of Lemma 5

The following lemma will now bound the number of times arm $i^*$ will not be the empirically best arm.

**Lemma 5** (Time when $i^*$ is not the anchor). *The total number of rounds when the best arm $i^*$ will not be the empirically best arm, even when it is in the active set, is upper bounded as*

$$\mathbf{E}\left[\sum_{r=1}^{T} \mathbb{1}[a_r \neq i^*, i^* \in A_r]\right] \leq \sum_{i \in [n] \setminus \{i^*\}} \frac{1}{\exp\{d(1/2, P_{i^*i}^{\mathrm{GCC}})\} - 1} \,,$$

*where $P_{i^*i}^{\mathrm{GCC}}$ is defined in Equation D.2.*

*Proof.* In the following we overload notation slightly and for a round $r$ define $N_{ii^*}(r)$ and $\hat{P}_{ii^*}(r)$ to be the equal to

$N_{ii^*}(t)$ and $\hat{P}_{ii^*}(t)$, where $t$ is the last trial in round $r$. We have the following set of inequalities:

$$\mathbf{E}\left[\sum_{r=1}^{T}\mathbb{1}[a_r \neq i^*, i^* \in A_r]\right] = \mathbf{E}\left[\sum_{r=1}^{T}\mathbb{1}[\exists i \neq i^*, i^* \in A_r, N_{ii^*}(r) > N_{ii^*}(r-1), \hat{P}_{i^*i}(r-1) \leq \frac{1}{2}]\right]$$

$$\leq \mathbf{E}\left[\sum_{r=1}^{T}\sum_{i \in [n]\setminus\{i^*\}}\mathbb{1}[i^* \in A_r, N_{ii^*}(r) > N_{ii^*}(r-1), \hat{P}_{i^*i}(r-1) \leq \frac{1}{2}]\right]$$

$$\leq \mathbf{E}\left[\sum_{r=1}^{T}\sum_{i \in [n]\setminus\{i^*\}}\sum_{n_i=0}^{T}\mathbb{1}[N_{ii^*}(r-1) = n_i, N_{ii^*}(r) > n_i, \hat{P}_{i^*i}^{n_i} \leq \frac{1}{2}]\right]$$

$$= \mathbf{E}\left[\sum_{i \in [n]\setminus\{i^*\}}\sum_{r=1}^{T}\sum_{n_i=0}^{T}\mathbb{1}[N_{ii^*}(r-1) = n_i, N_{ii^*}(r) > n_i, \hat{P}_{i^*i}^{n_i} \leq \frac{1}{2}]\right]$$

$$\leq \mathbf{E}\left[\sum_{i \in [n]\setminus\{i^*\}}\sum_{n_i=0}^{T}\mathbb{1}[\hat{P}_{i^*i}^{n_i} \leq \frac{1}{2}]\right]$$

$$= \sum_{i \in [n]\setminus\{i^*\}}\sum_{n_i=0}^{T}\mathbf{E}\left[\mathbb{1}[\hat{P}_{i^*i}^{n_i} \leq \frac{1}{2}]\right]$$

$$= \sum_{i \in [n]\setminus\{i^*\}}\sum_{n_i=0}^{T}\exp\{-n_i d(1/2, P_{i^*i}^{\mathrm{GCC}})\} \qquad \text{(using concentration Lemma 2)}$$

$$= \sum_{i \in [n]\setminus\{i^*\}}\frac{1}{\exp\{d(1/2, P_{i^*i}^{\mathrm{GCC}})\} - 1}$$

$\square$

### E.3   Proof of Lemma 6

In the next lemma we will bound the regret for the number of times an arm other than the best arm will be played.

**Lemma 6** (Regre due to a bad arm). *Given an arm $i \in [n] \setminus \{i^*\}$ the expected regret incurred due to arm $i$ when arm $i^*$ is the anchor is upper bounded as*

$$\mathbf{E}\left[\sum_{t=1}^{T}r(S_t, i) \cdot \mathbb{1}[a_t = i^*, i \in S_t]\right] \leq \Delta_{i^*i}^{\mathrm{GCC}} \cdot \frac{2e}{e-1} \cdot \left(\frac{(1+\delta)\log(TnC)}{d(P_{i^*i}^{\mathrm{GCC}}, \frac{1}{2})} + \frac{1}{\Omega(\delta^2)}\right),$$

*where $\delta > 0$ is some constant, and $\Delta_{i^*i}^{\mathrm{GCC}} = \max_{S:|S| \leq k} \Delta_{i^*i|S}$.*

*Proof.* Fix a value $n_i \in \{0, \cdots, T\}$. We will first upper bound the following quantity

$$\mathbf{E}\left[\sum_{t'=1}^{T}r(S_{t'}, i) \cdot \mathbb{1}[a_{t'} = i^*, i \in S_{t'}, N_{ii^*}(t'-1) = n_i]\right]. \tag{E.1}$$

This quantity bounds the total regret until the time $N_{ii^*}$ remains equal to $n_i$. Now, $N_{ii^*}$ is incremented in trial $t'$ if either $i^*$ or $i$. Hence, $N_{ii^*}$ is incremented in trial $t'$ with probability $P_{i|S_{t'}} + P_{i^*|S_{t'}}$. The total regret incurred due to the playing $i$ in trial $t'$ is given by $P_{i^*|S_{t'}} - P_{i|S_{t'}}$. Let us define $c_{t'} := P_{i|S_{t'}} + P_{i^*|S_{t'}}$, and $p_{t'} := P_{i^*|S_{t'}} - P_{i|S_{t'}}$.

The quantity in Equation E.1 is upper bounded by the cost of an experiment described in Fact 1 where the probability of success of coin $t'$ is given by $p_{t'}$ and its cost is given by $c_{t'}$. Using Fact 1 we have that

$$\mathbf{E}\left[\sum_{t'=1}^{T} r(S_{t'}, i) \cdot \mathbb{1}[a_{t'} = i^*, i \in S_{t'}, N_{ii^*}(t'-1) = n_i]\right] \le \Delta_{i^*i}^{\text{GCC}} \cdot \frac{2e}{e-1}.$$

Also, let $n_i^{\text{suf}} = \frac{(1+\delta)\log(TnC)}{d(P_{i^*i}^{\text{GCC}}, \frac{1}{2})}$. We can now upper bound the regret due to arm $i$ as

$$\mathbf{E}\left[\sum_{t=1}^{T} r(S_t, i) \cdot \mathbb{1}[a_t = i^*, i \in S_t]\right] = \mathbf{E}\left[\sum_{n_i=0}^{T}\sum_{t=1}^{T} r(S_t, i) \cdot \mathbb{1}[a_t = i^*, i \in S_t, N_{ii^*}(t-1) = n_i]\right]$$

$$\le \sum_{n_i=0}^{n_i^{\text{suf}}} \mathbf{E}\left[\sum_{t=1}^{T} r(S_t, i) \cdot \mathbb{1}[a_t = i^*, i \in S_t, N_{ii^*}(t-1) = n_i]\right]$$

$$+ \sum_{n_i=n_i^{\text{suf}}+1}^{T} \mathbf{E}\left[\sum_{t=1}^{T} r(S_t, i) \cdot \mathbb{1}[a_t = i^*, i \in S_t, N_{ii^*}(t-1) = n_i]\right]$$

$$\le n_i^{\text{suf}} \cdot \Delta_{i^*i}^{\text{GCC}} \cdot \frac{2e}{e-1}$$

$$+ \sum_{n_i=n_i^{\text{suf}}+1}^{T} \mathbf{E}\left[\sum_{t=1}^{T} r(S_t, i) \cdot \mathbb{1}[a_t = i^*, i \in S_t, N_{ii^*}(t-1) = n_i]\right]$$

We will now bound the second quantity in the above equation. Fix $n_i \in \{0, 1, \cdots, T\}$. Let $t' \in [T]$ be such that the event $\mathbb{1}[a_{t'} = i^*, i \in S_{t'}, N_{ii^*}(t'-1) = n_i - 1, N_{ii^*}(t') = n_i]$ holds if such a $t'$ exists, otherwise let $t' = T+1$. We have

$$\mathbf{E}\left[\sum_{t=1}^{T} r(S_t, i) \cdot \mathbb{1}[a_t = i^*, i \in S_t, N_{ii^*}(t-1) = n_i]\right]$$

$$= \mathbf{E}\left[\sum_{t=1}^{T} r(S_t, i) \cdot \mathbb{1}[a_t = i^*, i \in S_t, N_{ii^*}(t-1) = n_i, n_i \cdot d(\hat{P}_{ii^*}(t-1), \frac{1}{2}) \le \log(t-1) + \log(nC)]\right]$$

$$= \mathbf{E}\left[\sum_{t=1}^{T} r(S_t, i) \cdot \mathbb{1}[a_t = i^*, i \in S_t, N_{ii^*}(t-1) = n_i, n_i \cdot d(\hat{P}_{ii^*}(t-1), \frac{1}{2}) \le \log(t-1) + \log(nC)]\right]$$

$$= \mathbf{E}\left[\mathbb{1}[\exists t' \in [T] : N_{ii^*}(t') = n_i, \; n_i \cdot d(\hat{P}_{ii^*}^{n_i}, \frac{1}{2}) \le \log(t'nC)] \cdot \sum_{t=1}^{T} r(S_t, i) \cdot \mathbb{1}[a_t = i^*, i \in S_t, N_{ii^*}(t-1) = n_i]\right]$$

$$= \Pr\left[\exists t' \in [T] : N_{ii^*}(t') = n_i, \; n_i \cdot d(\hat{P}_{ii^*}^{n_i}, \frac{1}{2}) \le \log(t'nC)\right]$$

$$\cdot \mathbf{E}\left[\sum_{t=t'+1}^{T} r(S_t, i) \cdot \mathbb{1}[a_t = i^*, i \in S_t, N_{ii^*}(t-1) = n_i]\Big| \exists t' \in [T] : N_{ii^*}(t') = n_i, \; n_i \cdot d(\hat{P}_{ii^*}^{n_i}, \frac{1}{2}) \le \log(t'nC)\right]$$

We will bound the quantities in the above equation one by one. Using a similar argument as above and Fact 1 we have that

$$\mathbf{E}\left[\sum_{t=t'+1}^{T} r(S_t, i) \cdot \mathbb{1}[a_t = i^*, i \in S_t, N_{ii^*}(t-1) = n_i]\Big| \exists t' \in [T] : N_{ii^*}(t') = n_i, \; n_i \cdot d(\hat{P}_{ii^*}^{n_i}, \frac{1}{2}) \le \log(t'nC)\right] \le \Delta_{i^*i}^{\text{GCC}} \cdot \frac{2e}{e-1}.$$

This holds because of the fact that conditioning does not effect that events that happen after trial $t' + 1$. We finally have

$$\Pr\left( \exists t \in [T] : N_{ii^*}(t) = n_i, \ n_i d(\hat{P}_{ii^*}(t), \frac{1}{2}) \leq \log(tnC) \right)$$

$$\leq \Pr\left( \exists t \in [T] : N_{ii^*}(t) = n_i, \ n_i d(\hat{P}_{ii^*}(t), \frac{1}{2}) \leq \log(TnC) \right) \qquad (n_i \geq \frac{(1+\delta)\log(TnC)}{d(P_{i^*i}^{\text{GCC}}, \frac{1}{2})})$$

$$\leq \Pr\left( \exists t \in [T] : N_{ii^*}(t) = n_i, \ d(\hat{P}_{ii^*}(t), \frac{1}{2}) \leq \frac{d(P_{i^*i}^{\text{GCC}}, \frac{1}{2})}{1+\delta} \right).$$

We will let $P \in (\frac{1}{2}, P_{i^*i}^{\text{GCC}})$ to be a real number such that $d(P, \frac{1}{2}) = \frac{d(P_{i^*i}^{\text{GCC}}, \frac{1}{2})}{1+\delta}$, and use the concentration bound proved in Lemma 2, so that the above inequality can be written

$$\Pr\left( \exists t \in [T] : N_{ii^*}(t) = n_i, \ d(\hat{P}_{ii^*}(t), \frac{1}{2}) \leq d(P, \frac{1}{2}) \right) = \Pr\left( \exists t \in [T] : N_{ii^*}(t) = n_i, \ \hat{P}_{ii^*}(t) \geq 1 - P \right)$$

$$\leq \exp\left( -d(P, P_{i^*,i}^{\text{GCC}}) \cdot n_i \right).$$

Hence, we will have that

$$\sum_{n_i = n_i^{\text{suf}}+1}^{T} \mathbf{E}\left[ \sum_{t=1}^{T} r(S_t, i) \mathbb{1}[a_t = i, N_{ii^*}(t-1) = n_i, i^* \in S_t] \right] \leq \sum_{n_i = n_i^{\text{suf}}+1}^{} \Delta_{i^*i}^{\text{GCC}} \cdot \frac{2e}{e-1} \cdot \exp\left( -d(P, P_{i^*,i}^{\text{GCC}}) \cdot n_i \right)$$

$$\leq \Delta_{i^*i}^{\text{GCC}} \cdot \frac{2e}{e-1} \cdot \frac{1}{\exp\left( d(P, P_{i^*i}^{\text{GCC}}) \right) - 1}$$

$$\leq \Delta_{i^*i}^{\text{GCC}} \cdot \frac{2e}{e-1} \cdot \frac{1}{\Omega(\delta^2)}.$$

Hence, we have proved an upper bound as

$$\sum_{n_i=0}^{T} \mathbf{E}\left[ \sum_{t=1}^{T} \mathbb{1}[a_t = i, N_{ii^*}(t) = n_i, i^* \in A_t] \right] \leq \Delta_{i^*i}^{\text{GCC}} \cdot \frac{2e}{e-1} \cdot \left( \frac{(1+\delta)\log(TnC)}{d(P_{i^*i}^{\text{GCC}}, \frac{1}{2})} + \frac{1}{\Omega(\delta^2)} \right)$$

$\square$

# F Additional Information About Experimental Setup

## F.1 Synthetic Datasets

In this section we provide additional information about our synthetic datasets.

- **MNL-Exp:** A MNL model was generated by drawing random weights from the exponential distribution with parameter $\lambda = 3.5$, i.e. for arm $i \in [n]$, $\log v_i$ was sampled i.i.d. from $\text{Exp}(\lambda = 3.5)$.

- **MNL-Geom:** A MNL model was generated with weights $v_1 = e$, $v_2 = e^{\frac{1}{2}}$, ..., $v_n = e^{1/2^{n-1}}$.

- **GCC-One:** For this choice model, we selected arm 1 to be the GCW, and for each set $S$ containing arm 1, we set $p_{1|S} = 0.51$ and $p_{i|S} = \frac{0.49}{|S|-1} \ \forall i \in S \setminus \{1\}$; for sets $S$ not containing the GCW 1, we selected the smallest-index arm in $S$ to be the highest-probability arm $i_S^*$ in $S$, and set $p_{i_S^*|S} = 0.51$ and $p_{i|S} = \frac{0.49}{|S|-1} \ \forall i \in S \setminus \{i_S^*\}$).

- **GCC-Two:** For this choice model, we selected arm 1 to be the GCW, and for each set $S$ we defined $\Delta_S = \min\{\frac{|S|-1}{10}, 0.99\}$. If $i^* \notin S$ we selected the smallest-index arm in $S$ to be the highest-probability arm $i_S^*$ in $S$, otherwise we let $i_S^* := i^*$. We defined $P_{i_S^*|S} = \frac{1+\Delta_S}{|S|(1-\Delta_S)+2\Delta_S}$ and for any $i \in S \setminus \{i_S^*\}$, $P_{i|S} = \frac{1-\Delta_S}{|S|(1-\Delta_S)+2\Delta_S}$.

- **GCC-Three:** For this choice model, we selected arm 1 to be the GCW, and for each set $S$ we defined $\Delta_S = \max\{\frac{11-|S|}{11}, 0.01\}$. Given this definition of $\Delta_S$, the choice probabilities we defined in a similar manner as GCC-Two.

Note that the above GCC choice models are similar to the instance constructed in the proof of the lower bound, except that the $\Delta$ term now depends on the size of the set.

## F.2 Real-World Datasets

In this section we provide additional information for our real-world datasets.

**Estimation of choice models from real-world datasets.** We estimate choice probabilities from several real-world preference datasets, which contain multiple partial preference orders over items. The choice probability $P_{i|S}$ of an item $i$ over $S$, was taken to be the fraction of times in these partial order item $i$ was the top ranked items in $S$. More formally, let there be $m$ partial orders, $\mathcal{P}_1, \cdots, \mathcal{P}_m$, over $n$ items. For any subset $S \subseteq [n]$, and $i \in [n]$, let $N_{i|S}$ be defined as:

$$N_{i|S} := \sum_{j \in [m]} \mathbb{1}[\forall i' \in S \setminus \{i\} : i \succ_{\mathcal{P}_j} i'].$$

The choice probability $P_{i|S}$ is then estimated as:

$$P_{i|S} := \frac{N_{i|S}}{\sum_{i' \in S} N_{i'|S}}.$$

We conducted experiments on three real-world datasets.

- **Sushi:** This is a dataset from [24] which contains 5000 partial preference orders given by humans over 100 different types of sushis. Similar to [25], we selected a subset of 16 sushi types, such that there exists a GCW among them.

- **IrishMeath:** This is a dataset downloaded from *preflib.org* and contains data about elections held in Dublin, Ireland. The dataset contains $64,081$ partial preference orders given by humans over 14 candidates. We selected a subset of 12 candidates, such that there exists a GCW among them.

- **IrishDublin:** This dataset was also downloaded from *preflib.org* and also contains data about elections held in Dublin, Ireland. The dataset contains $29,988$ partial preference orders given by humans over 9 candidates. We again selected a subset of 8 candidates, such that there exists a GCW among them.

## F.3 Runtime and Space Complexity of WBA

The space complexity of our algorithm is $O(n^2)$ as it only stores the pairwise statistics extracted from multiway choices. Each trial of our algorithm runs in time polynomial in $n$. The most non-trivial step is computing $\mathcal{J}_i(t, C)$ for each arm. This step requires polynomial time because we can compute the quantity $\text{argmax}_{S \subseteq [n]} I_i(t, S) - |S| \cdot \log(nC)$ and check if it is greater than $\log(t)$. We compute $\text{argmax}_{S \subseteq [n]} I_i(t, S) - |S| \cdot \log(nC)$ by first sorting arms $j$ in the order of values $\mathbb{1}[\hat{P}_{ij}(t) \leq \frac{1}{2}] \cdot N_{ij}(t) \cdot d(\hat{P}_{ij}(t), \frac{1}{2})$. We then start with $S \leftarrow \emptyset$ and add one arm at a time from this sorted ordering to $S$. We stop adding arms to the set $S$ once the value $\mathbb{1}[\hat{P}_{ij}(t) \leq \frac{1}{2}] \cdot N_{ij}(t) \cdot d(\hat{P}_{ij}(t), \frac{1}{2})$ of the current arm $j$ is less than $\log(nC)$. It is easy to see that computing $I_i(t, S) - |S| \cdot \log(nC)$ for this set $S$ gives the value of $\text{argmax}_{S \subseteq [n]} I_i(t, S) - |S| \cdot \log(nC)$.

## F.4 Hardware Specifications

We ran all our experiments on a 32 core machine with Intel(R) Xeon(R) CPU E5-2683 v4 @ 2.10GHz processor cores. No GPUs were used in the experiments.

Figure 1: Dueling Bandit Regret ($R_{\text{DB}}$) defined in Appendix G v/s trials for our algorithm WBA (for $k = 2$) against dueling bandit algorithms (DTS, BTM, RUCB and RMED1) (the shaded region corresponds to std. deviation). As can be observed, our algorithm is competitive against these algorithms.

Figure 2: Dueling Bandit Regret ($R_{\text{DB}}$) defined in Appendix G v/s trials for our algorithm WBA against the MaxMinUCB (MMU) algorithm for $k = 2$ and $k = 5$ (the shaded region corresponds to std. deviation). We observe that our algorithm is better than MaxMinUCB on all datasets for both values of $k$. We further observe that under several datasets the regret achieved by our algorithm for $k > 2$ is better than the regret of our algorithm for $k = 2$.

# G   Results for Additional Notion of Regret

In the section we define a simple generalization of dueling bandit regret. All our results can be be extended to this notion of regret. Under this notion the regret for an arm is measured as the shortfall in the preference probability in a direct pairwise comparison to the best arm $i^*$.

**Definition 1.** *For a set $S \subseteq [n]$, we define the regret $r_{\mathrm{DB}}(S)$ to be*

$$r_{\mathrm{DB}}(S) = \sum_{i \in S} \left( P_{i^*|\{i,i^*\}} - P_{i|\{i,i^*\}} \right) . \tag{G.1}$$

This notion of regret allows for a more direct comparison between the regret of a choice bandits algorithm and a dueling bandits algorithm, as the regret for pulling an arm $i$ does not depend on the other arms pulled together with $i$. Using the definition of GCW $i^*$, it is easy to observe that $r_{\mathrm{DB}}(\{i^*\}) = 0$, and $0 \leq r_{\mathrm{DB}}(S) \leq |S|$ for any set $S \subseteq [n]$.

We present additional experimental results for this notion of regret. Figure 1 contains plots for comparisons with the dueling bandit algorithms, and Figure 2 contains plots for the comparisons of our algorithm with the MaxMinUCB algorithm. The experimental setup was the same as the one described in Section 6 and Appendix F. The overall conclusion with these experiments match the conclusions drawn from the experiments given in Section 6.

# H   Technical Fact

**Fact 1.** *Consider the following experiment: we repeatedly toss (independent) coins from a finite set $S$ of coins with different biases until we get a heads. Let the probability of heads for the $i$-th coin toss be given by $p_i \geq 0$, and the cost be given by $c_i$. The expected cost of this experiment is upper bounded as*

$$\mathbf{E}\left[ \sum_{i=1}^{|S|} \mathbb{1}[\textit{no heads till } i-1] \cdot c_i \right] \leq \frac{2c}{p} \cdot \frac{e}{e-1} ,$$

*where $\frac{c}{p} := \max_{i \in S} \frac{c_i}{p_i}$*

*Proof.* We will group the sequence of coin tosses such that each group has a total probability mass of at least 1. Formally, group $G_1$ will consist of the first $l_1$ coins such that $\sum_{i=1}^{l_1} p_i \geq 1$ and $l_1$ is minimized, group $G_2$ will consist of the next $l_2$ coins such that $\sum_{i=l_1+1}^{l_2} p_i \geq 1$ and $l_2$ is minimized, and so on. The probability that we do not see a head in the first group $G_1$ is upper bounded as

$$\prod_{i=1}^{l_1}(1-p_i) \leq \prod_{i=1}^{l_1} e^{-p_i} = e^{-\sum_{i=1}^{l_1} p_i} \leq e^{-1} .$$

A similar calculation works for each group, showing that we will see a success in a particular group with probability at least $1 - 1/e$.

Now, the amount of cost required for each group $c_G := \sum_{i \in G} c_i$ is upper bounded by $2 \max_{i \in S} \frac{c}{p}$. This is due to the fact that each group contains a probability mass of at most 2; and the fact that the maximum cost per a probability mass of $p$ is at most $c$, hence, the maximum cost per a probability mass of 2 can be at most $2c/p$.

$$\begin{aligned}
\mathbf{E}\left[ \sum_{i=1}^{\infty} \mathbb{1}[\text{no heads till } i-1] \cdot c_i \right] &= \mathbf{E}\left[ \sum_{j=1}^{\infty} \mathbb{1}[\text{no heads in group } G_{j-1}] \cdot c_{G_j} \right] \\
&\leq \frac{2c}{p} \cdot (1 - \frac{1}{e}) + \frac{4c}{p} \cdot \frac{1}{e} \cdot (1 - \frac{1}{e}) + \frac{6c}{p} \cdot \frac{1}{e^2} \cdot (1 - \frac{1}{e}) \cdots \\
&\leq \frac{2c}{p} \cdot \frac{e}{e-1} .
\end{aligned}$$

□