[Reviews · NeurIPS 2020]

Review 1

Summary and Contributions: The authors consider a generalization of duelling bandits where, at each time t, a learner selects a subset S_t of items S_t \subset \{1,...,n \} of size k < n or less, and the envinronment picks one of the items of S_t. The item picked most often is the preferred item of S_t, and we assume that there exists a Condorcet winner, i.e. an item which is preferred over any other item. For k=2 this is simply duelling bandits, for k > 2 this is a generalization of the problem. The authors prove a O(n^2 ln n + n (ln T)/Delta) regret lower bound, and they propse an algorithm which is order optimal, so that its regret is upper bounded by an expression of the same order.

Strengths: The problem is interesting, the proposed algorithm is both novel and order optimal, and the authors perfom numerical experiments to demonstrate that their approach is superior to the current state of the art. The paper is well written and easy to follow.

Weaknesses: - The statement of Thereom 1 seems imprecise. Is \Delta simply a parameter, or is it related to the reward gap for all models (I understand that it is linked to the reward gap for the MNL model, but for other models this seems unclear). If Delta is just a parameter, this would imply that the proposed algorithm cannot be claimed to be order optimal outside of the MNL model, contrary to what the authors seem to claim. - For this type of general structured bandit problems, general regret lower bounds are known (see for instance http://papers.nips.cc/paper/6773-minimal-exploration-in-structured-stochastic-bandits ), so that it is a pity that the authors did not investigate these. - The proposed algorithm has an input parameter C, which must be chosen "large enough" for the regret bound of Theorem 2 to hold. However the authors never specify what information is necessary to make this choice. Must C be greater than some universal constant ? Must it be chosen in a problem specific manner ? If so this greatly diminishes the value of the proposed algorithm, unless one has some sort of method to select C automatically.

Correctness: Yes

Clarity: Yes

Relation to Prior Work: Yes

Reproducibility: Yes

Additional Feedback:


Review 2

Summary and Contributions: The paper introduces the choice bandit problem, a natural generalization of the dueling bandits problem that has attracted significant interest from researchers over the last few years. In particular, the paper studies a problem where the learner is allowed to play more than k (>= 2) arms and receives the feedback in the form of which arm "wins" among the selected set of arms. Here, the learner is interested in identifying the "best arm" that has the highest probability of winning among any subset of arms. Assuming a fairly generic random utility framework for choice models (the mechanism by which winning arm is decided), the paper 1. establishes a fundamental lower bound for this problem and 2. presents an algorithm that achieves near optimal performance (with respect to the regret notion defined in the paper).

Strengths: 1. The problem considered in the paper, online learning in choice models is a an important problem with growing number of real world use cases. One particular strength of the paper is that they present an algorithm (and analysis) that is applicable beyond the multinomial logit choice model (MNL) whose limitations are well known to be used in practice. 2. The paper presents theoretical guarantees on the algorithm presented and backs it up by showing a fundamental lower bound on the problem.

Weaknesses: Though the paper is largely well written, there is room for improvement in the technical presentation of the paper. In particular, the paper as written is not completely self-containing (see below for more details) and a general reader might find it hard to understand all the details without reading the previous work.

Correctness: Yes.

Clarity: The theoretical notions (and the motivation) considered in the paper are not self-containing. In particular, the paper relies a lot more than usual on existing literature to justify many algorithmic and benchmark choices. For example, after reading the paper, I'm not sure about the high level goal of the paper, is it to identify one best arm or play assortments with many arms close to best arm? Related to this point, the notion of benchmark and regret also needs more description. There's a single line that this is motivated from the previous work. However, there's minimal justification on why this is the right notion of regret and what it aims to achieve. It is clear regret for playing S_t = i* is 0, but what happens when we play arms whose p(i|S) is much lower? For example, in the MNL model, including items whose v_i ~ 0 shouldn't hurt us if the goal is to ensure the best arm gets maximum chance to win. It is ideal to elaborate on the reasons why this notion of regret is in line with what they want. Lastly, I'd also like to see the paper motivating the problem from a real application point of view as well.

Relation to Prior Work: Yes

Reproducibility: Yes

Additional Feedback: Missing reference with respect to dynamic learning in the MNL model: Mnl-bandit: A dynamic learning approach to assortment selection. Post rebuttal: Acknowledging that I've read the author's response, other reviewers comment and happy to maintain the original score.


Review 3

Summary and Contributions: The paper deals with a variant of the dueling or battling bandit problem, where it is allowed to pull up to k many arms in each time step. In addition, a more general class of multiwise comparison models is considered as the ones investigated in the battling bandit problem or the previously used Multinomial Logit model of related settings. A lower bound on the expected regret is shown for the general class of multiwise comparison models, which in particular gives a more refined bound as the previously shown for the case of MNL models. Moreover, one learning algorithm is suggested for the considered problem and analyzed theoretically with respect to its expected regret bound. Finally, the proposed algorithm is investigated in numerous experiments on synthetic as well as real-world datasets and compared to related algorithms. ==================== Post Rebuttal ==================== After reading the other reviews as well as the author's response, I think that the authors have not addressed two of the concerns mentioned in the reviews thoroughly enough, namely the independence of k for the complexity terms as well as the suggestion to compare with the lower bounds coming from the structured bandits. In particular, I think it should be possible to recover a lower bound for their setting from the structured bandits, as the lower bound for the dueling bandit problem with Condorcet Winner can be recovered as well. However, as the authors give an own proof for their lower bound, I think this is appropriate. Nevertheless, it would have been interesting to include the corresponding OSSB algorithm for their problem in the experiments. The response regarding the independence of k for the complexity terms is somehow unsatisfactory, as it hasn't become clear what the "more subtle dependence" is. I really would have liked to see more details on this issue, for instance, an enlightening example. On the other side, the remaining issues such as the hidden dependencies of some terms in the upper bound, the high-level goal and the choice of C are addressed in an appropriate way and if the authors make the promised adaptations, I will be fine with that. All things considered, I would keep my original score.

Strengths: The paper is dealing with an interesting as well as practical relevant variant of the dueling/battling bandit problem. The multiwise comparison models considered in the paper are more general as the ones investigated in the battling bandit problem and also go beyond the previously considered Multinomial Logit model of some related works. Lower bounds on the expected regret for these general choice models are shown. The theoretical analysis is, except for some little things, sound and the paper is easy to read. The suggested algorithm is shown to be empirically appealing by comparison with existing methods, especially in the MNL case and moreover the theoretical guarantees for the MNL case are strict.

Weaknesses: I think that some relevant terms are missing in the regret upper bound (please see the detailed comments below), which will slightly reduce the attractiveness of the resulting upper bounds. Also, it seems that there is a gap between the upper and lower bounds for the general GCW model case.

Correctness: As already mentioned, there seems to be some missing terms in the regret upper bound. Also some statements in the discussion following Theorem 2 are questionable or speculative (please see the detailed comments below).

Clarity: The paper is well-written and easy to read. The main contributions are stated clearly. However, the authors are not specifying the admissible range for their parameter of the algorithm in order to obtain the stated regret upper bound and are unspecific with one relevant problem parameter in their lower bound result (please see the detailed comments below).

Relation to Prior Work: Yes, as the authors provide a paragraph on related work and also have an extensive section in the supplementary material.

Reproducibility: Yes

Additional Feedback: % Regret lower bound: The statement of the lower bound is peculiar as the role of the parameter \Delta is unclear. How does it relate to the underlying GCW problem instance? This relationship is of utmost importance in order to compare the lower bound result with the upper bound result. As far as I can tell, \Delta here is equivalent to \Delta_min^{GCC} of the underlying GCW problem, right? % Regret upper bound: I think the stated upper bound is hiding important terms in the log-term. More precisely, in your proof of Theorem 2 you are dropping at one point the factor "n times C "in the rightmost log-term. However, in my opinion, this is not entirely justified, since the parameter C has to be chosen larger than C' as defined on page 11, which in turn is large if P_{i^*,i}^GCC is near 1/2 or equivalently d(P_{i^*,i}^GCC,1/2) is near 0. Thus, d(P_{i^*,i}^GCC,1/2) is yet another hardness parameter occurring in the regret upper bound just as the already considered \Delta terms. By the way, each O-term in the proof of Theorem 2 should be O(n^2 log(n)). % Role of the parameter C: Related to the previous issue, it would be relevant to be more explicit on the smallest possible choice for the parameter C in the statement of Theorem 2, as this choice is depending on the relevant hardness parameters of the underlying problem instance as mentioned in the previous point. % Discussion after Theorem 2: If my conjecture regarding the \Delta term of the lower bound is correct, then the statement that the regret upper bound is asymptotically optimal is only valid for the case of the MNL model. In other words, there is a gap between the upper and lower bounds for the general GCW model case. Also the statement "It is also important to note that our regret bound does not depend directly on the choice set size k" is questionable. Both hardness parameter \Delta_min^{GCC} and \Delta_max^{GCC} depend on k and in particular lead to increasing regret bounds if k increases. Only if you could show that the ratio of \Delta_min^{GCC} and \Delta_max^{GCC} is bounded by a constant independent of k for any admissible k, you could infer the statement above. % Other little things: Main paper: - line 205: 'anchor arm a_t' - line 242: Delete "the" - line 272: measurable "set" Supplement: - Some of the long mathematical displays miss a period at the end. - Preselection Bandits also use a different notion of regret. Also it would make sense to mention them in the main part and include them in Table 1. - The weights w_i are defined incorrectly on page 5, as it should be exp instead of log. Also \kappa is not introduced. - In the proof of Lemma 2, there are various S's instead of S_t's. Moreover, Y_t has nothing to do with the event y_t=i^* and there should be Z_C = 1 for Y_t = i^* and Z_C = 0 for Y_t = i. - "Regret" after Lemma 4 on page 13. - n_i^suf is not an integer in general, so that you need to an appropriate rounding in the sums, which occur in the proof of Lemma 4. - Don't you need a union bound for "\exists t \in [T]" at the passage, where Lemma 2 is applied? If not you could be more precise here. - One real-world dataset should be the "IrishDublin" dataset on page 16. - The sum in the proof of Fact 1 is infinite, while you only have a finite set of coins according to the assertion.

[Author Response · NeurIPS 2020]

## Reviewer #1

**Re. statement of lower bound (Theorem 1):** The theorem statement should say "there exists a GCC choice model with $\Delta_{\min}^{\text{GCC}} = \Delta$ such that...". We will correct this in the final version if accepted.

**Re. structured bandits lower bound:** Thanks for pointing us to this reference (which we will certainly include). Although our choice bandits problem can be cast as a structured bandit problem, this existing lower bound is in terms of the solution of an LP, and it is not clear how this solution relates to the gaps in the underlying choice model instance. The main novelty in our lower bound is to construct a distribution over hard instances which allows us to identify a fundamental gap parameter quantifying the regret any algorithm must incur.

**Re. choice of parameter $C$ in our algorithm (and Theorem 2):** The parameter $C$ can indeed be chosen in a problem-independent manner. In particular, setting $C = T^4$ suffices for Theorem 2 to hold, giving a regret upper bound of $O(\log(TC)) = O(\log(T^5)) = O(\log(T))$. (If $T$ is not known, we can use the doubling trick.) To see why the choice $C = T^4$ suffices, note that in the proof of Theorem 2, we mention it is sufficient for $C$ to be larger than a term $C'$ (defined at the end of page 11 in the supplementary material). While the term $C'$ is problem-dependent, we do not actually need to know its exact value to set $C$ larger than it. In particular, note that $C'$ is upper bounded by $\left(1/\Delta_{\min}^{\text{GCC}}\right)^4$; moreover, in order to obtain any non-trivial upper bound for our algorithm, $\Delta_{\min}^{\text{GCC}}$ has to be larger than $1/T$. Hence, either $C'$ is upper bounded by $T^4$, or the instance is too hard to allow any non-trivial upper bound. Therefore, setting $C = T^4$ suffices to ensure $C \geq C'$ whenever the instance is not already too hard. We actually believe setting $C = T^4$ may be somewhat pessimistic (it arises from taking a union bound over all possible states of the algorithm in Lemma 2) – indeed, in our experiments, we set $C = 1$ for all datasets, and our algorithm still demonstrates sublinear regret with this choice – but it certainly suffices, and the regret bound with $C = T^4$ is at most a constant factor 5 times what one might get with $C = 1$ if the regret bound holds in that case. We will add a discussion on this in the final version if accepted.

## Reviewer #2

Thanks for the suggestions for making the paper more self-contained, and also for the reference on dynamic learning in the MNL model. We will incorporate all these suggestions and include this reference in the final version if accepted.

**Re. high-level goal (and notion of regret):** The goal is to identify the 'best' arm *while also playing good/competitive assortments during the exploration phase*. This is similar to dueling bandit settings where the goal is to identify the best arm while also playing good/competitive pairs during the exploration phase; we are *not* working in the pure exploration setting, where all assortments/pairs in the exploration phase incur uniform cost. Accordingly, our notion of regret penalizes an algorithm for playing assortments that are not 'competitive'. (E.g. in the case of the MNL model, if $v_i \approx 0$ for some arm $i$, then an algorithm incurs a lot of regret for playing $i$.) We will certainly clarify this.

**Re. real-world applications:** Essentially, we are interested in applications where the goal is to identify the 'best' item/product (or more generally, in future work, a collection of 'good' items/products), *while also serving well the users with whom the learning system interacts during the exploration phase in order to identify such items*. Applications could include online advertising, recommender systems, and online ranker evaluation for information retrieval. The latter can be viewed as a generalization of the ranker evaluation application considered by Yue and Joachims (ICML 2009), in which the goal was to identify the better of two ranking systems by interleaving their results and using a dueling bandit formulation; our setting would allow extending this to the identification of the best among $k \geq 2$ ranking systems by 'multi-leaving' their results, all while still presenting acceptable/good results to the users who are using the system during the exploration phase. We will be happy to add some comments on this as well.

## Reviewer #3

**Re. lower bound (Theorem 1)**: Yes, your intuition is correct. Please also see our response to Reviewer #1 above.

**Re. upper bound (Theorem 2):** You are right; the $O(n^2 \log n)$ term suppresses some problem dependent terms. The only reason for this was to simplify the exposition and to emphasize our algorithm is asymptotically order optimal as this term does not increase with $T$. The precise value of this term is given by $O(n \log n \times \sum_{i \neq i^*} \frac{1}{d(1/2, P_{i^*i}^{\text{GCC}})})$, where $d(\cdot, \cdot)$ is the KL-divergence. We have seen a similar approach to writing regret upper bounds in other papers (e.g. RMED paper), but we agree it would be good to be more precise upfront; we will certainly add a note on this.

**Re. parameter $C$:** This can be chosen in a problem-independent manner. Please see our response to Reviewer #1.

**Re. discussion after Theorem 2:** Our intention was to highlight that the bound is not *directly* a function of $k$, but rather has a more subtle dependence through the problem parameters. When $k$ increases, we consider larger sets and a broader problem instance, and hence, the terms $\Delta_{\max}^{\text{GCC}}$ and $\Delta_{\min}^{\text{GCC}}$ change. There is indeed a gap in the upper and lower bounds in the general GCC case, but even in this case, the dependence on other parameters such as $n$ and $T$ is asymptotically order-optimal. We will clarify these points in the final version if accepted.

[Meta-Review · NeurIPS 2020]

We thank the authors for their rebuttal. The latter did not address all issues raised by the reviewers (e.g. problem-specific lower bound and connection to structured bandits). However, all appreciated the novelty of the model and of the algorithm. We recommend the paper for acceptance and encourage the authors to account for the reviewers’ comments when preparing the camera-ready version of the paper.